# IceDetectNet: A rotated object detection algorithm for classifying components of aggregated ice crystals with a multi-label classification scheme

Huiying Zhang[1], Xia Li[2], Fabiola Ramelli[1], Robert O David[3], Julie Pasquier[1,4], and Jan Henneberger[1]

[1]Institute for Atmospheric and Climate Science, ETH Zurich, Zurich, Switzerland
[2]Institute for Machine Learning, ETH Zurich, Zurich, Switzerland
[3]Department of Geosciences, University of Oslo, Oslo, Norway
[4]Meteomatics AG, St. Gallen, Switzerland

**Correspondence:** Huiying Zhang (huiying.zhang@env.ethz.ch) and Xia Li (xia.li@inf.ethz.ch)

**Abstract.** The shape of ice crystals affects their radiative properties, growth rate, fall speed, and collision efficiency and thus, plays a significant role in cloud optical properties and precipitation formation. Ambient conditions like temperature and humidity determine the basic habit of ice crystals, while microphysical processes such as riming and aggregation further shape them, resulting in a diverse set of ice crystal shapes and effective densities. Current classification algorithms face two major challenges: (1) ice crystals are often classified as a whole (on the image scale), necessitating identification of the dominant component of aggregated ice crystals, and (2) single-label classifications lead to information loss because of the compromise between basic habit and microphysical process information. To address these limitations, here we present a two-pronged solution: a rotated object detection algorithm (IceDetectNet) that classifies each component of an aggregated ice crystal individually, and a multi-label classification scheme that considers both basic habits and physical processes simultaneously. IceDetectNet was trained and tested on two independent datasets obtained by a holographic imager during the NASCENT campaign in Ny-Ålesund, Svalbard, in November 2019 and April 2020. The algorithm correctly classifies 92 % of the ice crystals as either aggregate or non-aggregate and achieved an overall accuracy of 86 % for basic habits and 82 % for microphysical processes classification. On the component scale, IceDetectNet demonstrated high detection and classification accuracy across all sizes, indicating its ability to effectively classify individual components of aggregated ice crystals. Furthermore, the algorithm demonstrated good generalization ability by classifying ice crystals from an independent generalization dataset with overall accuracies above 70 %. IceDetectNet can provide a deeper understanding of ice crystal shapes, leading to better estimates of ice crystal mass, fall velocity, and radiative properties and thus, has the potential to improve precipitation forecasts and climate projections.

## 1 Introduction

The shape of ice crystals within clouds impacts the Earth's radiation budget (Ehrlich et al., 2008; Sun and Shine, 1994; Matus and L'Ecuyer, 2017; Flanner et al., 2007; Järvinen et al., 2018). As ice crystals interact with solar and terrestrial radiation they scatter, absorb and emit radiation, thereby influencing the radiative properties of the atmosphere (Flanner et al., 2007; Järvinen

et al., 2018; Yang et al., 2015). Furthermore, the shape of ice crystals has substantial effects on global precipitation, influencing both the spatial distribution and precipitation rate (Sterzinger and Igel, 2021; Woods et al., 2007; Jensen et al., 2017). The growth mechanisms of ice crystals play a crucial role in precipitation formation (Wegener, 1911; Findeisen, 1938; Lohmann et al., 2016; Kalina and Puxbaum, 1994; Mosimann et al., 1993). The efficiency of these processes is largely determined by the ice crystal shape, further highlighting its importance (Heymsfield, 1972; Khvorostyanov and Curry, 2002; Bailey and Hallett, 2004; Mitchell, 1996; Mitchell et al., 1990; Wang and Ji, 2000).

The initial shape of an ice crystal, also known as its basic habit (e.g., column, plate), is governed by the ambient meteorological conditions such as temperature and supersaturation that it experiences during its initial diffusional growth phase (Libbrecht 2016; Bailey and Hallett 2004). The change in the ambient environment, such as in a convective system, leads to a complex basic habit such as columns on capped columns (CPCs) (observed by (Pasquier et al., 2023)). They are further shaped by microphysical processes including riming (i.e., supercooled cloud droplets collide and freeze on the ice crystal) and aggregation (i.e., individual ice crystals collide and stick together). This leads to a wide range of ice crystal shapes, sizes and densities, introducing considerable challenges in the systematic classification of ice crystals.

Early ice crystal classification techniques used simple features like edge complexity (Cunningham, 1978), circular deficiency (Rahman et al., 1981), the surface area and perimeter (Duroure et al., 1994), the complexity (combined several geometric features such as particle area and area ratio (Schmitt and Heymsfield, 2014) to classify the shape of ice crystals, but cannot distinguish between composite ice crystals, such as irregular, aggregates, or bullet rosettes. More advanced techniques, like ice crystal classification with principal component analysis (Lindqvist et al., 2012) and logistic regression (Praz et al., 2017), have been developed and have achieved 80-90 % accuracy but still require manual feature extraction (e.g., aspect ratio). Furthermore, these algorithms demonstrated limitations in their ability to perform effectively on different datasets, as their classification performance was strongly influenced by the characteristics of the training dataset (Bishop and Nasrabadi, 2006; Goodfellow et al., 2016), which is defined as the generalization ability of the models. This dependency requires significant adjustments to the optimal thresholds when these algorithms are applied to new, unseen datasets.

The emergence of convolutional neural networks (CNNs) as part of deep learning algorithms has brought significant improvements in the classification of ice crystal habits, with their capability for automated feature extraction (Li et al., 2021; Albawi et al., 2017; Touloupas et al., 2020). Although CNNs exhibit a remarkable capacity to recognize key aspects of images, they struggle when faced with complex ice crystals such as CPCs or aggregates consisting of different basic habits (Zhang, 2021). Furthermore, CNNs that are based on single-label classification schemes face the challenge of information loss when composite ice crystals are classified (Zhang, 2021; Xiao et al., 2019). For example, an aggregated column can only be labeled either as 'aggregate' or 'column,' which results in information loss of either the basic habit or the microphysical process. According to the study by Korolev et al. (1999), in Arctic clouds, pristine ice habits (ice crystals without undergoing any microphysical processes) account for only 3 % of the particles observed, which would result in losing a substantial fraction (97 %) of ice information regarding either basic habits or microphysical processes when implementing a single-label classification scheme if the ice habits are still recognizable. Moreover, in stratiform clouds, Korolev et al. (2000) found 84 % of the ice crystals are irregular ice which is everything except needles and dendrite. These irregular ice crystals would be either

aged or aggregated by our definition (refer Sect. 2 and Sect. B). To tackle this problem, Zhang et al. (2022) first proposed that the ice shape label should contain two types of information: basic habits and the microphysical processes it experienced, and thus a multi-label should be assigned to one ice. Jaffeux et al. (2022) combined data from the Precipitation Imaging Probe and 2DS-Stereo Probe to train CNNs to classify ice crystals according to their basic habit and occurrence of riming and aggregation. Although their study considered potential microphysical processes for each ice crystal category manually after the CNN classification, the specific microphysical processes associated with individual components of an aggregated ice crystal remained unknown.

To summarize, there are two key limitations of current ice crystal classification algorithms: (1) Algorithms often classify the images of an ice crystal as a whole, necessitating the identification of the dominant component of an aggregated ice crystal and thus, are not able to account for the presence of multiple basic habits in an aggregated ice crystal. (2) Single-label classification algorithms require a compromise between basic habit and microphysical process information, leading to information loss. To address these issues, we propose a novel approach that consists of a rotated object detection algorithm (called IceDetectNet) along with a multi-label classification scheme. IceDetectNet can classify the ice crystals down to the scale of aggregated ice crystal components with both basic habit and microphysical process information, thereby eliminating the need to identify a dominant aggregated component. The multi-label classification scheme simultaneously accounts for both basic habits and microphysical processes of an ice crystal, reducing information loss. However, like all supervised learning methods, our approach is limited to the ice categories present in the training dataset, limiting its applicability until the model is fine-tuned on a new dataset. The data used to train and test IceDetectNet are described in Sect. 2. The structure of IceDetectNet is presented in Sect. 3. The performance of our proposed algorithm is evaluated in Sect. 4. Finally, Sect. 6 presents our conclusions and the relevant discussion of this study.

## 2    Data description

The data used in this study was collected in Arctic mixed-phase clouds during the NASCENT campaign (Pasquier et al., 2022a) conducted in Ny-Ålesund, Norway. Ice crystal images were captured by the holographic imager HOLIMO mounted on the tethered balloon system HoloBalloon (Ramelli et al., 2020). The measured ice particle sizes ranged from 50 $\mu$m to 2.4 mm. First, the cloud particles were classified as cloud liquid droplets and ice crystals using a convolutional neural network (CNN) approach, as described in (Touloupas et al., 2020). This preliminary classification served as the basis for the subsequent detailed classification of the ice crystals.

Following this initial categorization, each ice crystal was classified into one of seven basic habits: 'column', 'plate', 'lollipop' (Pasquier et al., 2022a), 'CPC', 'irregular', 'frozen droplets', and 'small'. Our seven basic habit categories were determined by their presence and distinct shape features observed in our dataset collected in Arctic mixed-phase clouds in NyAlesund. These basic habit classes are based on the categories used in Pasquier et al. (2022a) as we used the same dataset. Additionally, up to two microphysical process attributes (i.e., 'aggregate' and 'aged') were assigned to each ice crystal. Table 1 describes the seven basic habits and four microphysical processes categories (i.e., 'pristine', 'aged', 'aggregate' and 'aged and aggregate').

Thus, the final habit classification of an ice crystal is a combination of the basic habit and microphysical processes (final classification = basic habit + microphysical processes). Not all combinations of basic habits and microphysical processes are feasible, resulting in a total of 19 ice classes (examples are shown in Fig. B1) rather than the theoretically possible 28 ($7 \times 4$) categories. For instance, the 'small' class refers ice crystals that are too small to determine their habit, making it impossible to derive their microphysical processing. Furthermore, all ice crystals in the classes 'lollipop', 'CPC' and 'irregular' are defined as aged ice instead of pristine ice (as they are not newly-produced ice) while they are still basic habit categories. For a more detailed discussion of the categorization criteria, we refer to Appendix B. However, due to data limitations, our dataset does not capture every possible basic ice habit, such as needles and rosettes, but the existence of these ice habits is well acknowledged (Kikuchi et al., 2013). This limitation is acknowledged and further discussed in Sect. 5 where we look at potential extensions to IceDetectNet. As new data containing additional ice habits become available, IceDetectNet can be updated as it is designed to incorporate these new habits through fine-tuning, ensuring the continued evolution of the model."

The dataset collected on November 11, 2019 (Pasquier et al., 2022b), hereafter training dataset, was used to train IceDetect-Net. During the training it was divided into a training subset (comprising 80 % of the data) and a validation subset (made up of the remaining 20 %), using a cross-validation method (detailed introduction in Sect. 3.8.1). This validation subset serves a similar purpose as the traditional test sets used in other studies ((Jaffeux et al., 2022; Xiao et al., 2019; Touloupas et al., 2020), providing an initial evaluation of the model's performance under known conditions. On the other hand, the generalization dataset was collected on a different date, April 1, 2020 (Pasquier et al., 2022b) which is not used during training but to evaluate the generalization abilities of IceDetectNet.

Figure B1 offers a summary of both training and generalization datasets. The training dataset consists of 18'864 ice particles, where the 'column' and 'CPC' were the dominant classes, accounting for 47.5 %. Non-pristine ice, which is ice that is not freshly formed, accounts for 70.5 % of the ice crystals in the training data. Additionally, 18.8 % of the non-pristine ice crystals have undergone two microphysical processes and aggregated ice makes up 12 % of the ice crystals in the training dataset.

In contrast, the generalization dataset has a significant fraction of 'irregular' (47.3 %) and 'small' (23.4 %) ice crystals (Fig. B1). Unlike the training dataset, the generalization dataset does not include any instances of the 'lollipop' or 'CPC' classes and, consequently, the corresponding compound categories 'lollipop-aggregate' or 'CPC-aggregate' do not exist. Moreover, the generalization dataset only contains three occurrences of the 'frozen droplet' class, with small numbers for 'frozen droplet-aged' (8) and 'frozen droplet-aged-aggregate' (12), while 'frozen droplet-aggregate' is not present. The proportion of non-pristine ice increases from 70.5 % in the training dataset to 93.9 % in the generalization dataset. The fraction of aggregate ice increases from 11.9 % in the training dataset to 37.7 % in the generalization dataset.

The difference between the training and generalization datasets is an example of the natural variability of field observations. In our case, the two datasets were collected during different seasons, resulting in variations in the environmental conditions. The training dataset was collected in the temperature range from -8 to -3 °C (mostly in the column regime) while the generalization dataset was collected between -23 and -15 °C (mostly in the plate regime) (Pasquier et al., 2022b). These differences allow us to assess the generalization ability of IceDetectNet and to examine its performance in diverse environmental conditions.

**Table 1.** Description of the ice crystal categories including seven basic habits and four microphysical process categories. All 'small' ice crystals are categorized as 'pristine' ('#'), while all the ice crystals in the classes marked with '*' are categorized as 'aged'.

| Property | Class | Description |
|---|---|---|
| **Basic habits** | Column | Columnar ice crystal |
| | Plate | Plate-like ice crystal |
| | Frozen droplet | Frozen cloud or drizzle droplets are characterized by non-spherical shapes or distortions. |
| | Small# | Ice crystals that seem small by the eye of the hand labelor (usually smaller than 75 μm). |
| | Columns on capped-columns (CPC) * | Ice crystals that contain both columnar and plate-like features, often resembling an 'H'. These ice crystals are formed when growing in both the column and plate temperature regimes (Pasquier et al., 2023). |
| | Lollipop* | Lollipop are formed when a supercooled droplet collides with columns and freezes upon impact (Keppas et al., 2017) |
| | Irregular* | Irregular-shaped ice crystal with no clearly defined ice habit. |
| **Microphysical processes** | Pristine | Ice crystals with an easily identifiable shape that have not undergone any microphysical processes. |
| | Aged | Ice crystals that have undergone microphysical processes such as riming or sublimation. |
| | Aggregate | Ice particles that are composed of two or more ice crystals stuck together. |
| | Aged & aggregate | Ice crystals that have undergone both aging and aggregation. |

## 3 Methodology

In this section, we first provide an overview of Convolutional Neural Networks (CNNs), which serve as the foundation for the object detection algorithm in IceDetectNet. We start by explaining the overall structure of CNNs (Sect. 3.1), followed by an introduction to the rotated object detection algorithm developed in this study and implemented into IceDetectNet (Sect. 3.2). Subsequently, we discuss the data preparation (Sect. 3.4) and training process (Sect. 3.7), outlining the essential steps for training the model. Lastly, the evaluation metrics are introduced that are used to assess the performance of IceDetectNet on the training dataset and its ability to generalize to the unseen generalization dataset (Sect. 3.8).

| Class | Train | Generalization | Aggregate | Non-Pristine | Example |
|---|---|---|---|---|---|
| Column | 5442 | 329 | − | − | |
| Plate | 126 | 556 | − | − | |
| Frozen droplets | 201 | 3 | − | + | |
| Small | 1618 | 3393 | − | + | |
| Columns on capped-columns | 3509 | 0 | − | + | |
| Lollipop | 419 | 0 | − | + | |
| Irregular | 1106 | 3613 | − | + | |
| Column-aged | 3603 | 729 | − | + | |
| Plate-aged | 90 | 393 | − | + | |
| Frozen droplets-aged | 490 | 8 | − | + | |
| Column-aggregate | 17 | 44 | + | + | |
| Column-aged-aggregate | 794 | 1346 | + | + | |
| Plate-aggregate | 10 | 114 | + | + | |
| Plate-aged-aggregate | 90 | 711 | + | + | |
| Frozen droplets-aggregate | 65 | 0 | + | + | |
| Frozen droplets-aged-aggregate | 220 | 12 | + | + | |
| Irregular-aged-aggregate | 620 | 3239 | + | + | |
| Capped column-aged-aggregate | 409 | 0 | + | + | |
| Lollipop-aged-aggregate | 30 | 0 | + | + | |
| Overall | 18864 | 14490 | | | |

**Table 2.** Overview of the number of ice crystals in each class for both the training and generalization datasets, along with illustrative examples for each class. The green circles with a plus symbol indicate that the corresponding ice class is part of the Aggregate/non-Pristine category in the dataset, while the orange circles with a minus symbol indicate the opposite. Please note that the examples shown are intended for visual reference only and may not be representative of the entire dataset.

## 3.1 Convolutional Neural Networks (CNNs)

CNNs are a class of neural networks widely recognized for their exceptional performance in image classification tasks (Gu et al., 2018; Albawi et al., 2017; Rawat and Wang, 2017; Touloupas et al., 2020). CNNs consist of a specific architecture designed to extract meaningful features from images. The key components of a typical CNN include convolutional layers, pooling layers, and fully connected layers (Fig. 1). These layers work together to enable effective image analysis. The convolutional layer scans the input image with a small filter or kernel, extracting low-level features such as edges and color. The pooling layer reduces the spatial size of the convolved feature (Feature Map shown in Fig. 1) and aims to decrease computational complexity. The fully connected layer, which is usually the final layer of a CNN, performs the classification using the flattened or pooled output from the preceding layers.

In practice, CNN structures can be much more complex than the basic CNN described above. He et al. (2016) proposed a deep residual learning approach, which stores input information and propagates it directly from the first layer to the last. This approach has been successfully used in subsequent object detection algorithms that utilize the ResNet-50 structure (He et al., 2016). Due to this success, IceDetectNet (described in Sect. 3.5) is trained using the pre-existing parameters of ResNet-50, which was trained on the ImageNet dataset consisting of approximately 1.3 million images labeled into 1000 categories, with the exception of the last layer due to the different number of categories in ice classification (Deng et al., 2009; He et al., 2016). This helps to speed up the training process and achieve better performance.

## 3.2 Rotated object detection algorithm

Building on the foundations of CNNs, object detection algorithms serve as an extension to detect and classify objects within images. While CNNs typically classify the image as a whole, object detection algorithms localize and classify specific objects within these images, providing both their location (through the bounding box) and class labels (Zhao et al., 2019). The rectangular box that tightly encloses the object of interest is called a bounding box (as shown in Fig. 2). Rotated object detection algorithms additionally predict the angle of rotation of the bounding box (Zou et al., 2023). In this study, we introduce such a rotated object detection algorithm for ice crystal classification as part of IceDetectNet. This algorithm classifies multiple components of aggregated ice crystals individually and predicts the center, the dimensions and the rotation angle of bounding boxes enclosing the ice crystal components. This ensures that the ice components are captured within the smallest feasible rectangle, which offers a more accurate recognition of the object by minimizing the inclusion of background pixels (Ding et al., 2019). Here, we use the S2ANet network structure (Han et al., 2021) as the base structure of the rotated object detection algorithm within the IceDetectNet.

## 3.3 Hand-labeling of bounding boxes and ice categories

Accurate hand labeling is essential for training IceDetectNet, as is for all supervised learning methods. In contrast to conventional classification algorithms, rotated object detection models such as IceDetectNet require a dual process of hand labeling.

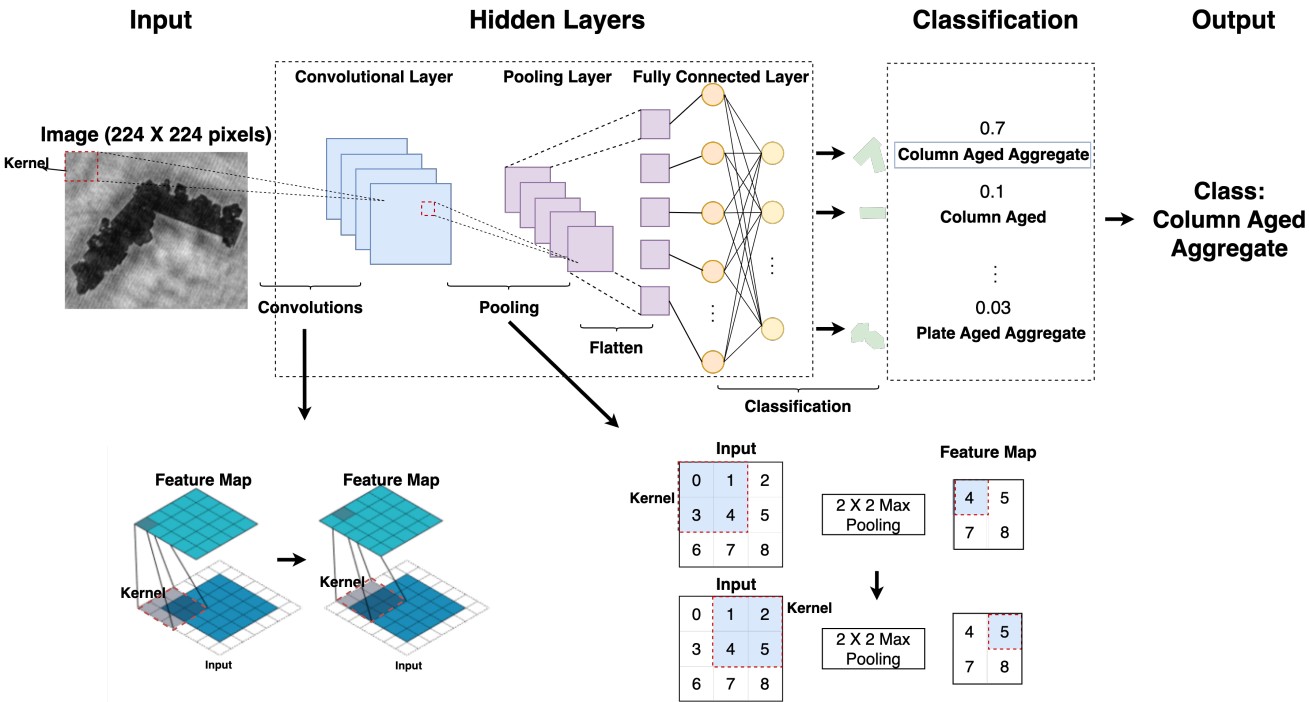

**Figure 1.** General structure of a CNN with an example of ice crystal classification. The input ice crystal is classified into one of the ice classes based on computed probabilities, with the highest probability determining the assigned class. In this example, the input ice crystal is classified as 'column-aged-aggregate' with a 70 % probability.

The first is to locate the ice components within the images by drawing bounding boxes and the second is to assign the appropriate category labels to each component based on our multi-label classification scheme.

In the present study, both the training and generalization datasets were initially hand-labeled on the image scale using our multi-label ice classification scheme (Sect. 2). For the hand-labeling on the image scale, the basic habit of the largest ice component is considered the basic habit of the image. Regarding microphysical processes, any image containing an ice component showing signs of aging was labeled 'aged'. Additionally, images consisting of multiple ice components were categorized as 'aggregated'. The hand-labeling on the image scale served as the basis for the hand-labeling on the component

scale.

For hand-labeling of non-aggregated ice crystals (identified using the hand label on the image scale), we applied an automated method for drawing bounding boxes. The method uses the color contrast between the typically black ice pixels and the typically grey background to identify the location of the ice component regions. Then we calculated the minimum bounding rectangle of the ice regions automatically as the bounding box of these non-aggregated ice crystals. The non-aggregated ice

crystals were assigned the same ice category labels as the corresponding labels on the image scale.

For aggregated ice crystals, the hand-labeling of the bounding boxes (i.e., drawing) was done manually using the platform provided by hub.ango.ai as illustrated in Appendix A and Fig. A1. Bounding boxes were manually drawn representing the minimum enclosing rectangle of each ice component. Furthermore, every bounding box was visually classified in an ice category following the multi-label classification scheme introduced in Sect. 2. In total, we manually labeled 2255 aggregated ice crystals and 16609 non-aggregated ice crystal components. Note that the hand-labeling on a component scale was only done for the training dataset due to the large effort involved with the hand-labeling of bounding boxes and ice categories. The generalization dataset was only hand-labelled on the image scale.

## 3.4 Image preprocessing

Before the image is fed into IceDetectNet, the initial image is enlarged by 15 % (in both length and width) to ensure full coverage of bounding box drawing and maintain the aspect ratio (see input in Fig. 2) by adding black pixels (pixel values = 1) around the borders. This augmentation ensures that the entire bounding box is located within the image, even when parts of the bounding box extend beyond the original image frame. To ensure consistency across the network training and testing, all images are then uniformly resized to $512 \times 512$ pixels by using bilinear interpolation after the enlargement.

The input images are normalized to meet the pre-trained ResNet-50 model's input specifications. For example, the pre-trained ResNet-50 model requires an RGB image as input, which consists of three dimensions by default. Given that our images only have one dimension, we replicate the single dimension three times to emulate the three-dimensional structure of RGB images and to produce pseudo-RGB images.

## 3.5 Inference process of IceDetectNet

The structure of the IceDetectNet algorithm is shown in Fig. 2. The input for IceDetectNet (Fig 2 step 1) are the processed images undergoing the preprocessing steps described in Sect. 3.4. The algorithm uses the ResNet-50 backbone network (He et al., 2016) to extract image features on a per-pixel scale. After feature extraction, IceDetectNet predicts potential bounding boxes for individual ice components (Fig. 2 step 2). These predicted bounding boxes contain the location, size, and rotation angle of the respective ice components. Multiple bounding boxes might be predicted for the same ice component, or some predicted bounding boxes may be too large to tightly capture an ice component, while others could be too small, missing some parts of an ice component (Fig. 2 step 2). Prior to classification, duplicate bounding boxes are removed (Fig. 2 step 3) by a feature alignment module (Sect. 3.6). The remaining predicted bounding boxes enclosing the individual ice components are the input for the classification module (Fig. 2 step 4), which outputs a predicted ice label with a confidence level (Fig. 2 step 5). After classification, a post-processing step is performed to further remove duplicate bounding boxes (Fig. 2 step 6) by comparing all predicted bounding boxes instead of hand-labeled bounding boxes using the Intersection over Union (IoU) threshold and the confidence level of classification. The IoU quantifies the overlap between two predicted bounding boxes and is calculated as:

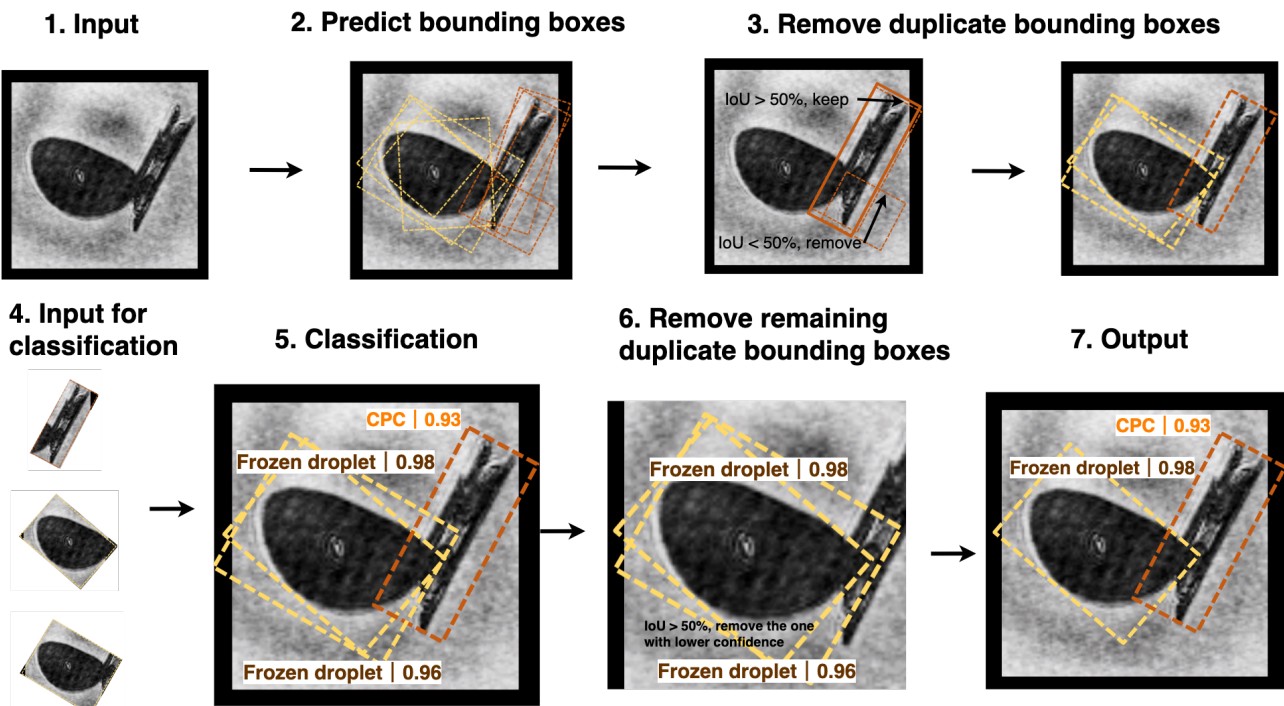

**Figure 2.** Structure of the IceDetectNet algorithm consisting of predicting potential bounding boxes (step 2), removing duplicate predicted bounding boxes (step 3), cropping the remaining bounding box for classification (step 4) and predicting the ice crystal categories of each bounding box (step 5). The yellow and orange dashed lines indicate bounding boxes predicted by the algorithm and their corresponding labels and confidence intervals (step 5), whereas the solid lines show the hand-labeled bounding boxes (step 3). The individual steps are described in the main text.

$$\mathrm{IoU} \; = \; \frac{\text{Area of overlap}}{\text{Area of union}} \tag{1}$$

If the IoU between two predicted bounding boxes exceeds a threshold (in our case, IoU > 50 %), the bounding box with the lower confidence level is discarded. This threshold was set to 50 % in the present study to minimize the number of misclassified aggregated ice crystals as non-aggregated and vice versa. Previous studies suggested that an IoU threshold within the range of 50 % to 75 % leads to the best performance (Zhang et al., 2019).

### 3.6 Training process of IceDetectNet

The training phase of IceDetectNet is a crucial and complex process because it requires a careful balance between reducing detection errors (Fig. 2 step 2) and classification errors (Fig. 2 step 5), all through a process of loss minimization. The feature

alignment module (Fig. 2 step 3) is trained to reduce the difference between the predicted and hand-labeled bounding boxes. If the IoU between a predicted and a hand-labeled bounding box is above 50 %, the prediction is considered correct (Zhang et al., 2019). On the contrary, an IoU below 50 % indicates an incorrect prediction, resulting in a loss that the training process then aims to minimize. To reduce the loss, a method called backpropagation is employed. Backpropagation adjusts the model's parameters to reduce errors and refines the feature extraction to improve the accuracy of bounding box prediction in step 2. The primary objective of the feature alignment module is to fine-tune the orientation and positions of bounding boxes, especially those with an IoU below 50 %. The images within the refined bounding boxes are then processed by the classification module. Incorrectly predicted labels contribute to the model's loss in classification, which is further minimized using backpropagation as well, leading to improved classification in step 5. Steps 6 and 7 belong to post-processing and are not subject to training.

### 3.7 Training details

The training details of IceDetectNet and the hyperparameters used during training are described here. To prevent the model from memorizing the training data (a problem known as 'overfitting') and to lower the generalization errors when it is applied to new unseen data, we introduced transformations to our training images. More specifically, we applied a technique called data augmentation which performs random flips of the images in horizontal, vertical, and diagonal directions with a 25 % probability. During the inference, no data augmentation was applied to prevent any distortion in the final output.

The training was executed on a computational system equipped with four RTX 2080 GPUs. A batch size of 64 was chosen to optimize computational efficiency and training stability.

The learning rate underwent a structured adaptation during the training process as follows:

1. Initially, the learning rate was set to 0 and linearly increased to 0.0025 over the initial 500 steps. Here, a 'step' is defined as a single iteration in the training process, in which one batch of data is processed to update IceDetectNet's parameters.

2. After the first 500 steps, the learning rate was kept constant at 0.0025.

3. A reduction by a factor of 10 was applied at specific epochs: the learning rate was set to 0.00025 from the 64th to the 88th epochs and further reduced to 0.000025 after the 88th epoch. This decremental strategy aimed at refining model parameters with progressively smaller updates as the training advanced.

To ensure model robustness and prevent overfitting, we employed the early stopping technique (Jabbar and Khan, 2015). Checkpoints were integrated to retain the best-performing model based on the validation dataset.

### 3.8 Evaluation

To evaluate the performance of IceDetectNet, we used a cross-validation approach (Sect. 3.8.1) and a range of evaluation metrics (Sect. 3.8.2): overall accuracy, precision, recall, and the confusion matrix, as highlighted in the following sections.

### 3.8.1 Cross-validation

IceDetectNet was validated by applying a five-fold cross-validation approach, which is a method that minimizes the variance in performance estimation while maximizing the use of available data for training (Browne, 2000). Here we randomly partitioned the dataset into five equally sized sub-samples or 'folds'. Four of the five sub-samples were used for training the model while the remaining sub-sample was retained as the validation data for testing the model. This process was then repeated five times (the folds), with each of the five sub-samples used exactly once for validation. Each fold was designed to include images from every class, thus ensuring that the model was trained and evaluated on a diverse set of ice crystals from all categories. This helped to prevent bias in the evaluation of the model's performance due to an unrepresentative selection of training and test data (Arlot and Celisse, 2010). The five outcomes from the folds were then averaged to produce a single estimation of IceDetectNet's performance.

In future applications, if the performances of the five individual models are similar (performance are evaluated in Sect. 4), a single model could be selected for use, simplifying the process. Alternatively, the ensemble of five models can be used to make the final predictions, increasing reliability. For example, if 3 models predict one ice component as a 'column' while 2 predict it as a 'plate', then the component is predicted as a 'column'. The advantage of this method over repeated random subsampling is that all ice crystals are used for both training and validation and each ice crystal is used for validation exactly once. This method, though computationally expensive, provides a robust evaluation of IceDetectNet's performance and its ability to generalize to new, unseen data (Arlot and Celisse, 2010).

### 3.8.2 Evaluation Metrics

To assess the performance of IceDetectNet, we employ several metrics that evaluate the model performance with regard to different aspects, including overall accuracy, precision, recall, confusion matrix and F1 score. The overall accuracy is defined as the ratio of the number of correct predictions to the total number of particles (Goodfellow et al., 2016). An overall accuracy of 100 % means that, for example, all ice particles were correctly predicted, while an overall accuracy of 0 % indicates that all particles were mispredicted. While overall accuracy provides a quick and straightforward metric to interpret the model performance, it can be misleading when dealing with imbalanced datasets where classes are not equally represented. In such cases, the model may perform well in predicting the dominant classes but struggle with predicting rare classes. Precision and recall both measure the accuracy of a deep-learning classification model in predicting a single category from two perspectives. Precision is calculated as the ratio of the number of correct predictions of a specific class to the total number of predictions (Goodfellow et al., 2016), while recall is defined by the ratio of the number of correct predictions of a specific class to the total number of this class (Goodfellow et al., 2016). A high precision score indicates effective identification of a specific class, while a high recall score indicates that the model excels in identifying instances of a particular class and is less likely to miss relevant instances that belong to the class. All of these metrics can be combined and visualized in a so-called confusion matrix (Goodfellow et al., 2016). In a confusion matrix, the diagonal, from top-left to bottom-right, corresponds to correct predictions made by the model, while the elements outside this diagonal represent misclassifications. The bottom-right cell of the matrix

displays the total number of ice crystals and the overall accuracy. The bottom row provides the actual counts per class and their respective per-class precision. Similarly, the rightmost column presents the predicted counts per class and the associated per-class recall. The F1 score is a harmonized metric that combines precision and recall, providing a balanced measure of a model's performance, particularly in situations where the balance between precision and recall is critical (Goodfellow et al., 2016). This score reaches its best value at 1 (indicating perfect precision and recall) and its worst value at 0. In the context of IceDetectNet, a high F1 score would indicate not only that the model accurately identifies ice particles (high precision), but also that it successfully detects the majority of actual ice particles (high recall), making it a robust metric for evaluating model performance across different classes, especially in the presence of imbalanced datasets.

## 4   Results

### 4.1   Evaluation of model performance

The evaluation of IceDetectNet differs from traditional deep learning classification algorithms because both detection (Sect. 4.1.1) and classification (Sect. 4.1.2) steps need to be evaluated. As described in Sect. 3.8.1, we trained five models using a five-fold cross-validation approach. Four folds were used for training, and the remaining fold was used for validation. A small portion of the images for which no bounding boxes were predicted (validation fold in training dataset: 11/3755) were labeled as 'none' and excluded from the following analysis.

### 4.1.1   Performance of aggregate detection

The detection performance of IceDetectNet was examined by first evaluating the ability of the algorithm to distinguish between aggregate and non-aggregate ice. Images with a single bounding box were defined as non-aggregate ice, whereas images with multiple bounding boxes were defined as aggregate ice. The aggregate/non-aggregate detection was evaluated by comparing the number of predicted bounding boxes with the number of hand-labeled bounding boxes for the training dataset (Fig. 3a). Hand-labeled and predicted bounding boxes are in good agreement, with an overall accuracy of 92 %, reflecting the ability of IceDetectNet to correctly classify images as aggregated or non-aggregated ice.

To understand the source of the 8 % misdetected aggregate and non-aggregate ice, we analyzed the number of overdetected and underdetected bounding boxes. Here we consider it overdetection when the algorithm predicts multiple bounding boxes for an ice crystal that is hand-labeled as non-aggregate (i.e. one bounding box) and underdetection when the algorithm predicts one bounding box for an ice crystal that is hand-labeled as aggregate (i.e. multiple bounding boxes). In absolute numbers, there were 266 instances of overdetection and 63 instances of underdetection (Fig. 3b). In relative terms, 40 % of the predicted aggregates were hand-labeled as non-aggregates (overdetection), resulting in a recall of 60 % for actual non-aggregates. Conversely, only 2 % of the predicted non-aggregates were hand-labeled as aggregates (underdetection), indicating a high recall of 98 % for actual aggregates (Fig. 3c). This shows that while the model tends to overestimate the presence of aggregates, it is highly effective at identifying actual aggregates. Considering that only 12 % of the training dataset was aggregate ice (as detailed in

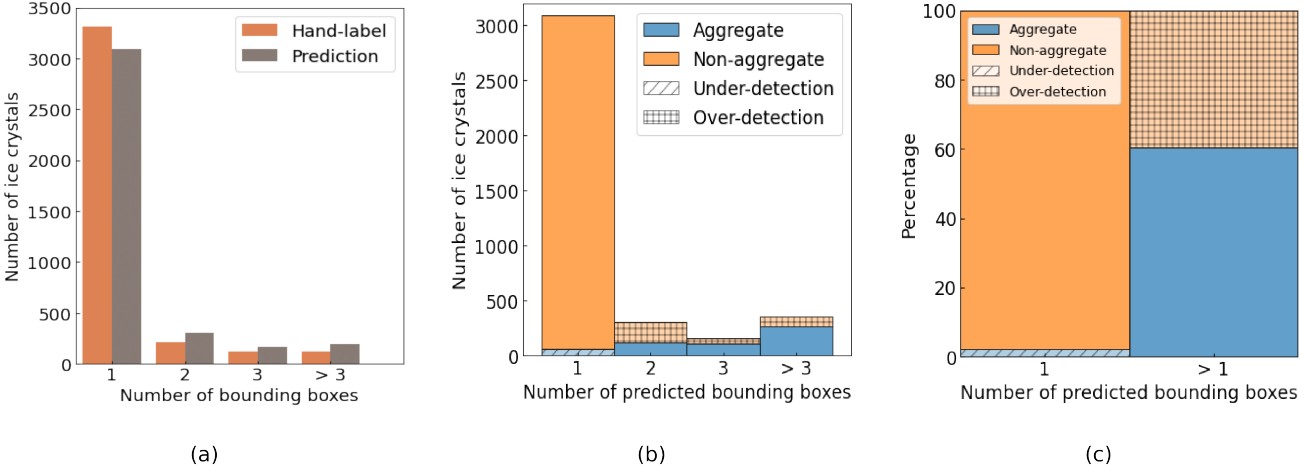

**Figure 3.** (a) Histogram of the number of hand-labeled (orange) and model-predicted (brown) bounding boxes (average over five models). Instances with more than three bounding boxes are combined into a single category and instances with zero detected bounding boxes (11) are excluded. (b) Histogram with the number of predicted bounding boxes for hand-labeled aggregated (blue) and non-aggregated (orange) ice crystals. The shaded orange and blue regions denote areas of overdetection and underdetection, respectively. (c) Same as panel (b), but all bounding boxes larger than one are combined, thereby providing an intuitive visualization of the percentages of overdetection and underdetection.

Sect. 2), a few mispredictions of non-aggregate ice as aggregate can significantly increase the overdetection. For example, if 2 % of the non-aggregate (66 ice crystals) were misclassified as aggregate, this would lead to a 14.6 % overdetection. Up to now, we evaluated how well the predicted categories match the actual categories. As a complement, we now turn our attention to precision, examining the accuracy of the predictions in terms of correctly identified categories. The precision is 88.7 % for aggregate (i.e., 88.7 % of the predicted aggregates were hand-labeled as aggregates) and 77.1 % for non-aggregate, consistently showing the model's tendency to over-predict the numbers of bounding boxes.

To address the issue of overdetection or underdetection of bounding boxes, it is possible to adjust the IoU threshold in the post-processing (as introduced in Sect. 3.2). In the present study, an IoU threshold of 50 % was applied to remove duplicate bounding boxes (see Fig. 2), but the IoU threshold can be changed based on the relative composition of the ice classes in the dataset. Generally, when an overdetection problem was identified, a higher IoU threshold could be implemented to reduce the number of detected bounding boxes, and the opposite adjustment could be made if underdetection was observed. Thus, the IoU threshold can be used as a tuning parameter to reduce/increase the number of bounding boxes kept after the post-processing.

**4.1.2 Performance of ice classification**

The classification performance of IceDetectNet was examined by quantifying the accuracy with which detected components are categorized into their respective basic habit and microphysical processes classes (following the multi-label classification

scheme detailed in Sect. 2). As discussed in Sect. 3.3, the basic habit of an ice crystal (i.e. image scale) is determined by the largest bounding box. The presence or absence of an 'aged' classification is based on the detection of aging signatures among all bounding boxes, while aggregation is defined by ice crystals with more than one bounding box. We evaluated the performance of the five trained models using the cross-validation approach described in Sect. 3.8.1. The overall accuracies for 5 folds and their mean values and the mean F1 scores for ice multi-label classification (19 classes), basic habit classification (7 classes), and microphysical process classification (4 classes) are shown in Table 3. The mean overall accuracies range between 78 % (for multi-label classification) and 86 % (for basic habit classification) and thus indicate good classification performance. Standard deviation values of OA below 1 % on all data demonstrate robust results among the five models. In contrast, the performance on average F1 scores, ranging from 54.9 %(for multi-label classification) to 78.8 % (for basic habit classification) is generally worse than on average OA, indicating that IceDetectNet performs worse on rare classes. Furthermore, the low standard deviation values of averaged F1 scores (below 2 %) on all data further indicate the 5 models have relatively the same results.

To gain further insights into IceDetectNet's performance in each ice category, we analyzed the confusion matrices (mean of 5 models) for basic habit classification (Fig. 4) and microphysical processes classification (Fig. 5). IceDetectNet achieved an overall accuracy of 86 % for the basic habit categories (Fig. 4). The confusion matrix shows that the IceDetectNet performed well for the ice categories that are represented as a large fraction in the dataset, like 'column' (precision of 90 %, 1978 instances) and 'small' (precision of 93 %, 324 instances). However, IceDetectNet encountered challenges in accurately classifying rare classes such as 'plate' (67 %, 47 instances) and 'lollipop' (77 %, 73 instances). The main source of misclassification for plates was confusion with 'column' (7 instances). For the microphysical processes category (Fig. 5, IceDetectNet achieved an overall accuracy of 82 %. While the model performed well in identifying 'Pristine' ice crystals (92 %), it showed lower performance in predicting 'aggregate' (47 %) and 'aged-aggregate' (49 %) ice crystals. This might be explained by the imbalanced dataset, where 'Pristine' ice crystals dominated with a total contribution of 66 %. This suggests that balancing the dataset could further optimize IceDetectNet's classification performance for 'aggregate' and 'aged-aggregate' ice crystals in future iterations. A closer examination of the misclassified ice crystal images shows that the primary source of error was an underdetection of the number of bounding boxes. For example, in an image containing two aged columns, only one 'column-aged' crystal was detected resulting in mislabeling it as 'aged' instead of 'aged-aggregate'.

To investigate the classification performance of IceDetectNet in simpler scenarios, we evaluated the performance on non-aggregated ice and aggregated ice separately (Table 4). The evaluation on non-aggregated ice provides a benchmark because non-aggregated ice images consist of a single ice component and thus allows us to compare the performance to traditional deep learning algorithms. When considering only non-aggregated ice crystals, IceDetectNet has an accuracy of 82 % for all data, 90 % for basic habits, and 85 % for microphysical processes (Table 3). Previous studies using single-label classification (Xiao et al., 2019; Jaffeux et al., 2022; Zhang, 2021) have reported overall accuracies above 90 %, which is higher compared to IceDetectNet for all data (82 %). However, for the multi-label classification, IceDetectNet classifies both basic habits and microphysical processes. While non-aggregated ice does not have an aggregation process, aging processes are still present. When one considers solely the basic habit classification, the accuracy of IceDetectNet (90 %) for non-aggregated ice closely aligns

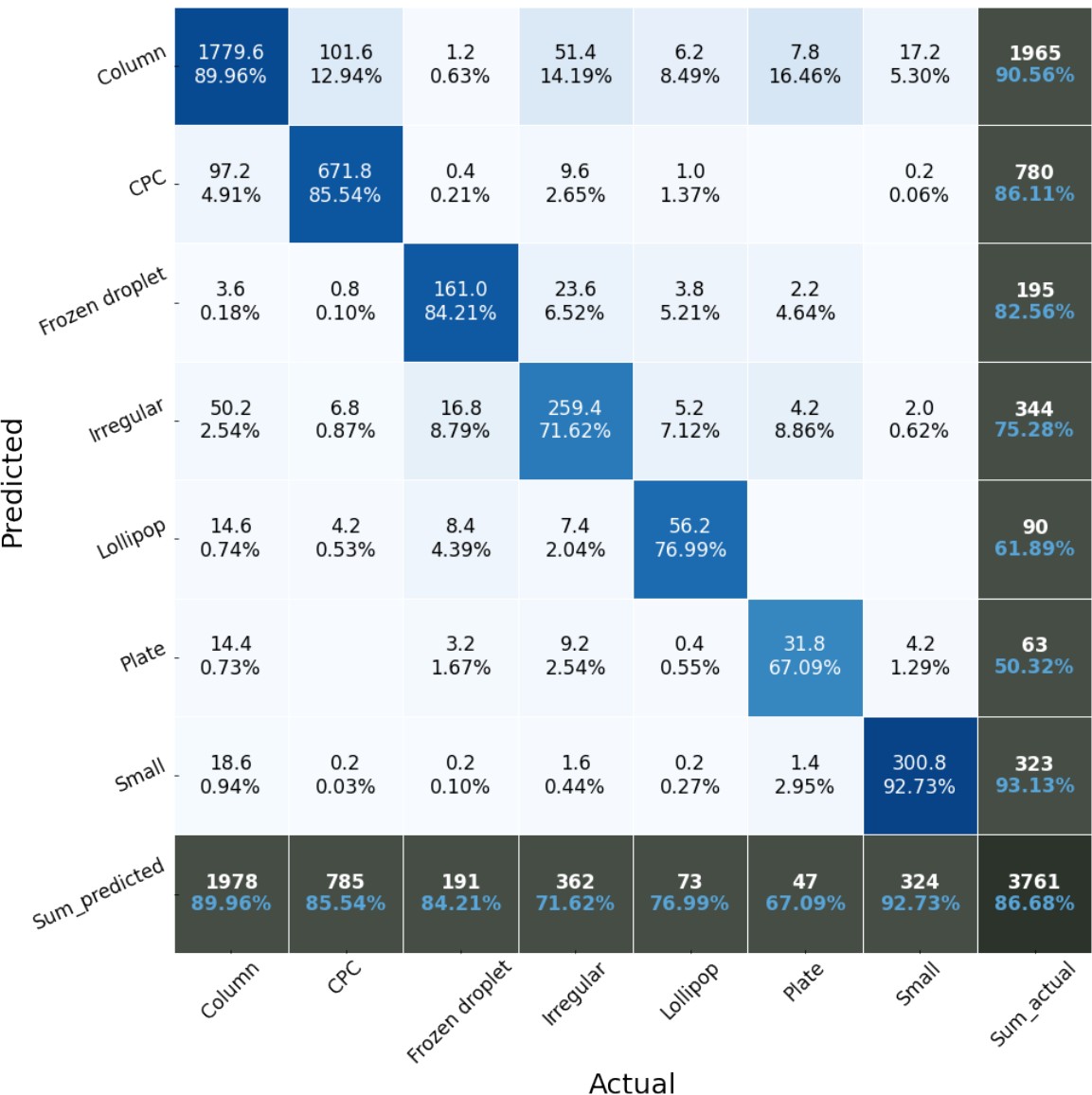

**Figure 4.** Confusion matrix of the mean performance of the basic habit classification for the training dataset (mean of 5 models). The y-axis represents predicted values, while the x-axis represents hand-labeled values. Bottom black row: The number of hand-labeled ice crystals (white) and precision (blue) in each class. The bottom right box shows the overall number of ice crystals (white) and the overall accuracy (blue). Rightmost black column: The number of ice crystals predicted (white) and recall (blue) in each class. The boxes in the middle (non-black boxes) evaluate the hand-labeled and predicted labels of the classification. For example, the second box in the first row means that 101.6 ice crystals are predicted as 'column' but the actual labels of these 101.6 ice crystals are 'columns on capped-columns' (CPCs). The percentage in this box represents the ratio of the number of ice crystals in this box (i.e. 101.6) to the total number of hand-labeled 'CPCs' (i.e. 785).

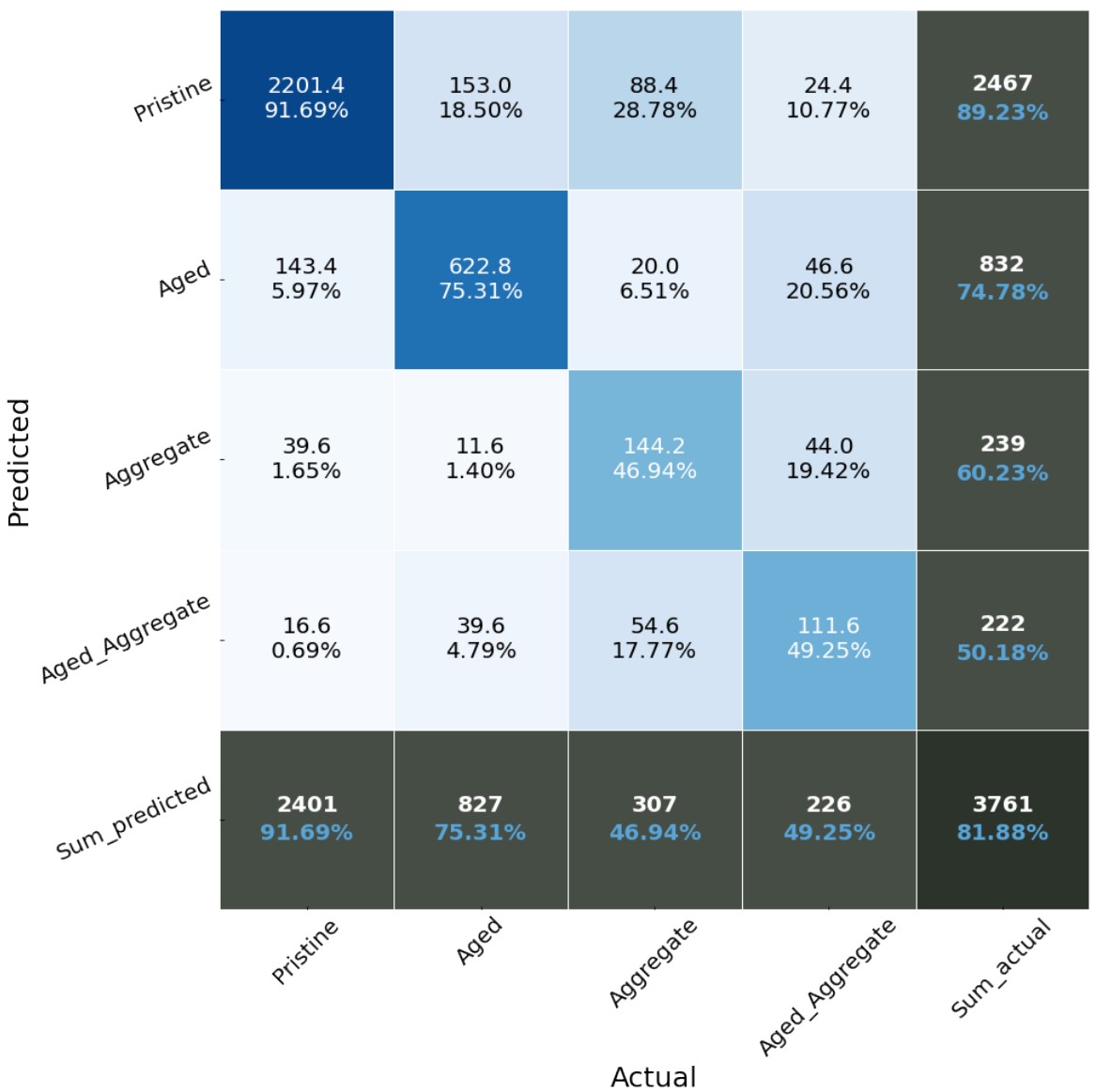

**Figure 5.** Similar confusion matrix as Fig. 4, but for physical processes.

**Table 3.** Overall accuracy of the multi-label, basic habit and microphysical processes ice classification. The table displays the overall accuracy values for each of the five models, along with the mean and standard deviation (std) values (all values are reported in percentages). The validation set is broken down into 'aggregate' (agg) and 'non-aggregate' (non-agg) subsets.

| | | 1 | 2 | 3 | 4 | 5 | mean OA | mean F1 score | std-OA | std-F1 |
|---|---|---|---|---|---|---|---|---|---|---|
| **Multi-label** | **All data (19-class)** | 78.1 | 78.0 | 78.3 | 77.0 | 79.4 | **78.2** | **54.9** | 0.9 | 1.9 |
| | Non-agg (10-class) | 82.5 | 81.3 | 81.3 | 83.5 | 82.0 | 82.1 | 71.7 | 0.9 | 1.6 |
| | Agg (9-class) | 46.1 | 53.7 | 54.4 | 47.0 | 50.5 | 50.3 | 41.4 | 3.8 | 5.9 |
| **Basic Habit** | **All data (7-class)** | 86.5 | 86.5 | 86.3 | 85.6 | 87.2 | **86.4** | **78.8** | 0.6 | 1.3 |
| | Non-agg (7-class) | 89.4 | 90.0 | 89.4 | 88.8 | 90.7 | 89.7 | 81.8 | 0.7 | 1.3 |
| | Agg (6-class) | 71.7 | 76.1 | 72.6 | 70.7 | 71.3 | 72.5 | 58.5 | 2.1 | 3.4 |
| **Microphysical Processes** | **All data (4-class)** | 81.4 | 82.0 | 81.3 | 80.8 | 82.6 | **81.6** | **66.9** | 0.7 | 1.1 |
| | Non-agg (2-class) | 85.6 | 84.0 | 84.6 | 84.7 | 86.2 | 85.0 | 84.3 | 0.9 | 1.0 |
| | Agg (2-class) | 48.9 | 55.0 | 56.1 | 48.3 | 52.4 | 52.1 | 62.3 | 3.5 | 2.8 |

with the results reported in the aforementioned studies. Thus, under the same classification domain, IceDetectNet performs competitively with existing classification models and offers additional information regarding microphysical processes.

When shifting our focus to aggregated ice, the inherent complexity of classifying the multiple components of an aggregate becomes evident by a decreased classification accuracy (Table 3). The accuracy drops to 50 % for multi-label, 72 % for basic habits, and 52 % for microphysical processes. The reduction in performance between non-aggregate and aggregate ice subsets was more pronounced for the classification of microphysical processes (85 % to 52 %), compared to the classification of the basic habit (90 % to 72 %). This suggests that the reduced performance of IceDetectNet in all-class classification can be

attributed primarily to the challenges in classifying microphysical processes.

## 4.2   Detection and classification performance on aggregated component scale

In the previous sections, the detection and classification performance of IceDetectNet was evaluated on an image scale. In this section, our focus shifts from the image scale to the aggregate component scale, specifically to the bounding boxes of aggregate ice crystals. To evaluate the detection and classification performance on an aggregate component level, we followed

a structured approach:

1. Pairing of bounding boxes: For each hand-labeled bounding box of an aggregate, we search for the predicted bounding box with the highest IoU.

2. Detection performance: The detection was considered correct if the IoU between the paired predicted and the hand-labeled bounding box was larger than 0.5, otherwise the detection was considered incorrect.

3. Classification performance: The classification was considered correct if the label of the hand-labeled bounding box matched the label of the paired predicted bounding box, otherwise it was considered incorrect.

In contrast to the previous sections where the mean performance of all five models was examined, the performance was evaluated on a single model which was randomly selected (due to the robustness among the model runs). We categorized the bounding boxes into 'small', 'medium', and 'large' using the areas of the predicted bounding boxes. The thresholds were set at the 33 % and 66 % percentiles of all bounding box areas, corresponding to below 32,331 pixels, from 32,331 to 71,275 pixels, and above 71,275 pixels, respectively. For a more intuitive understanding, these ranges correspond to squares with side lengths of below 180 pixels, from 180 to 267 pixels, and above 267 pixels, respectively.

When evaluating the detection and classification performance of IceDetectNet at the aggregate component scale for the three bounding box size categories (small, medium, large; Fig. 6), we find good detection performance among all size categories with accuracies ranging between 84 % (small bounding box) and 72 % (large bounding box). The detection performance decreases for larger bounding box sizes, which might be explained by an increased variability in appearance, texture and scale for larger bounding boxes. The classification accuracies for correctly detected bounding boxes ranged between 66 % and 71 %, with medium-sized boxes achieving the highest classification accuracy of 71 %. This suggests that medium boxes may offer an ideal balance between detectability and feature richness. Thus, IceDetectNet shows good detection and classification performance for bounding box sizes down to 662 pixels and up to 294,903 pixels.

In general, the detection performance of IceDetectNet on the aggregate component scale is higher (72 - 84 %) than the classification performance (66 - 71 %). When combining the detection and classification performances (i.e., detection × classification), similar overall performances as reported in Table 3 for the aggregate subset (i.e. image scale) are obtained (50 %). For example, the accuracy for small bounding boxes was determined to be 54.6 %, derived from the product of 65 % classification accuracy and 84 % detection accuracy. The higher performance in detection compared to classification suggests that the lower performance observed in the aggregate subset compared to the non-aggregate subset (as described in Sec. 4.1.2) is primarily due to misclassifications and not misdetections.

## 4.3 Generalization ability of IceDetectNet

### 4.3.1 Generalization ability on aggregate detection

The generalization ability of IceDetectNet was evaluated by applying it to an independent generalization dataset, which was not used to train the algorithm and was collected during a different season (detailed in Sect. 2). The same evaluation for detection and classification on the image scale was performed as for the training dataset (Sect. 4.1.1).

The detection performance for aggregates of the generalization dataset shows an overall accuracy of 84 % with an overdetection error of 26 % and an underdetection error of 8 % (Fig. 7a, b). This corresponds to a 14 % decrease in overdetection and a 6 % increase in underdetection compared to the training dataset. It is consistent across both the training (as described in Sec. 4.1.1) and generalization datasets that overdetection is a larger problem than underdetection. However, the different ice category distribution in the generalization dataset, specifically the increase in the aggregate from 12 % in the training dataset to 37.7 % in the generalization dataset, is likely responsible for the shift in underdetection and overdetection between the train-

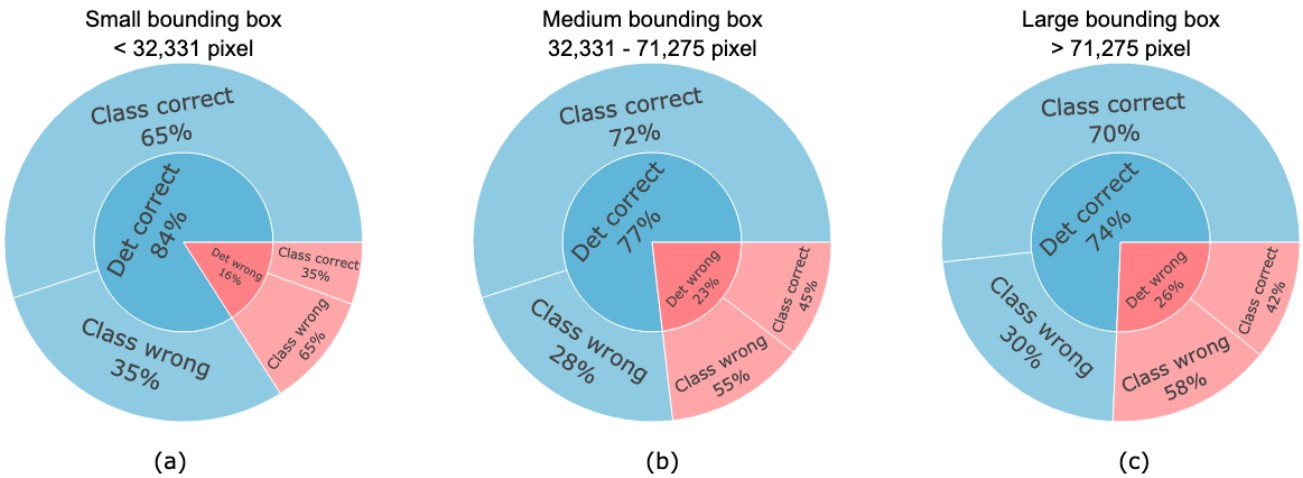

**Figure 6.** Sunburst diagrams to evaluate the detection (det) and classification (class) performance for (a) small, (b) medium, and (c) large bounding boxes sizes. Inner layers show detection results, while outer layers show classification results. The percentages indicate the proportion of bounding boxes in each category that were correctly or incorrectly detected and classified.

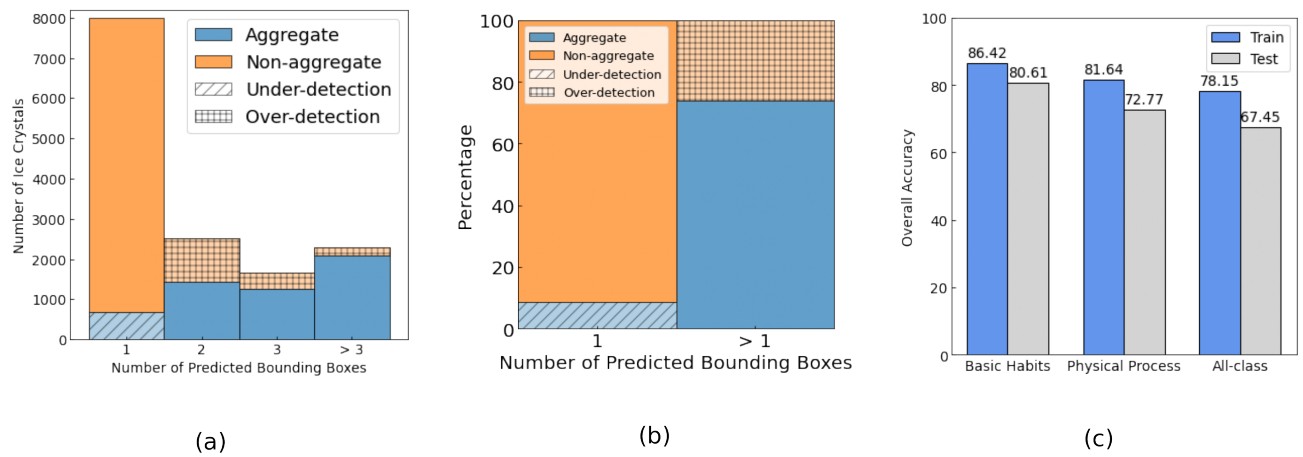

**Figure 7.** Panel (a) and (b) show the same as Fig. 3 (b) and (c), but for the generalization dataset. Panel (c) shows the overall accuracy of IceDetectNet for the training dataset (blue) and generalization dataset (grey) in classifying basic habits, microphysical processes, and all-classes.

ing and generalization datasets. Thus, when the algorithm is applied to the generalization dataset (with a higher fraction of aggregate), the problem of overdetection is reduced but still exists.

### 4.3.2 Generalization ability on ice classification

The classification accuracy on the generalization dataset showed overall accuracies ranging from 67 % (for the 'all data' category) to 80 % (for the 'basic habits' category) (see Fig. 7c and Table 4). As consistent as the performance on the training dataset, the values of OA are still higher than averaged F1 scores, ranging from 48.5 % (for the 'all data' category) to 68.7 % (for the 'basic habits' category) (see Table 4). Compared to the training dataset, the classification performance decreased. The most significant decrease of 11 %, was observed in the classification of 'all classes' on OA, while the smallest decrease was observed in the classification of 'basic habits' (6 %) on OA and the classification of 'microphysical processes' (2.1 %) on averaged F1 score. The observed decrease in performance between the training and generalization datasets is small, especially when considering domain shift, a situation where the data distribution differs between the training and generalization datasets (Stacke et al., 2020). Specifically, the training and generalization datasets have different compositions of non-pristine ice crystals, with 71 % in the training set and 94 % in the test set. IceDetectNet's ability to adapt to different data compositions and maintain relatively high accuracy indicates the ability of the algorithm to generalize to different dataset characteristics. Within the 'non-aggregate' subset, IceDetectNet showed good classification performance for 'all classes' (73 %), 'basic habits' (86 %), and 'microphysical processes' (78 %) on OA (as summarized in Table 4). Though the averaged F1 scores are still lower than OA, the decrease percentage in averaged F1 scores is smaller than in OA. In the 'aggregate' subset, performance levels were lower than in the 'non-aggregate' subsets (50-68 % on OA; 45-68 % on averaged F1 scores). The observed reduction in the aggregate subset is mainly due to domain shift, reflecting the difference in aggregate ice fraction between the training (12 %) and generalization (38 %) datasets. The generalization dataset, with a higher proportion of aggregated ice, presented more complexity and variability. Aggregates, with their multiple bounding boxes and variable structures, are inherently more difficult to classify than non-aggregates. This increased complexity in the test set likely contributed to the drop in performance, highlighting the challenges posed by domain shifts in the data. Consistently low standard deviation values (below 3 % on OA; below 2 % on averaged F1 scores) were observed across all models in all data subsets, which indicate the stable and reproducible performance of IceDetectNet. These standard deviation values represent the variability in classification performance among different runs or configurations of the IceDetectNet model.

To gain further insights into IceDetectNet's performance in each ice category, here we analyzed the confusion matrices (mean of 5 models) for basic habit classification (Fig. 8) and microphysical processes classification (Fig. 9) for the generalization dataset as well.

IceDetectNet achieved an overall accuracy of 81 % for the basic habit categories (Fig. 8) The confusion matrix shows that IceDetectNet still performed better for the ice categories that comprise a large fraction of the dataset, like 'small' (precision of 90 % ). However, among these, 'column' classification performance had a large performance drop (18 % decrease in precision compared to the training dataset), and its two main misprediction sources were 'irregular' and 'plate', which almost represent all the mispredictions (29 % ). This could be due to the data distribution shift from 'column' to 'plate'. Under the general decrease

**Table 4.** Overall accuracy of the multi-label, basic habit and microphysical processes ice classification. The table displays the overall accuracy values for each of the five models, along with the mean and standard deviation (std) values (all values are reported in percentages). The generalization dataset is broken down into 'aggregate' and 'non-aggregate' subsets.

| | | 1 | 2 | 3 | 4 | 5 | mean OA | mean F1 score | std-OA | std-F1 |
|---|---|---|---|---|---|---|---|---|---|---|
| | **All data (14-class)** | 67.5 | 66.7 | 67.2 | 67.5 | 68.3 | **67.5** | **48.5** | **0.6** | **1.6** |
| **Multi-label** | Non-agg (8-class) | 72.8 | 73.3 | 71.2 | 74.5 | 73.2 | 73.0 | 58.3 | 1.2 | 2.0 |
| | Agg (6-class) | 46.3 | 50.0 | 53.1 | 47.7 | 51.2 | 49.7 | 45.4 | 2.7 | 1.4 |
| | **All data (7-class)** | 81.6 | 79.9 | 79.6 | 80.7 | 81.3 | **80.6** | **68.7** | **0.8** | **0.6** |
| **Basic habit** | Non-agg (5-class) | 86.4 | 85.9 | 85.4 | 84.8 | 86.7 | 85.9 | 70.3 | 0.7 | 0.8 |
| | Agg (5-class) | 64.7 | 69.1 | 65.3 | 69.6 | 71.7 | 68.1 | 61.0 | 2.9 | 1.7 |
| | **All data (4-class)** | 72.8 | 72.3 | 72.5 | 72.9 | 73.4 | **72.77** | **64.8** | **0.4** | **0.3** |
| **Microphysical processes** | Non-agg (2-class) | 78.5 | 78.0 | 77.4 | 78.1 | 77.3 | 77.8 | 74.6 | 0.5 | 0.8 |
| | Agg (2-class) | 45.2 | 51.1 | 50.9 | 46.6 | 48.4 | 48.4 | 67.9 | 2.6 | 0.6 |

trend among all categories, 'irregular' surprisingly has a 10 % increase in precision. The main misprediction of 'irregular' comes from 'column' in both the training dataset and generalization data, which could be the reason that IceDetectNet learned many column features in the training dataset and thus distributed higher weights on these column features. While the number of 'column' is much less in the generalization dataset and thus leads to better performance in classifying 'irregular'. For the missing categories like 'lollipop' and 'CPC' that had zero actual occurrences, IceDetectNet still predicted 113 ice as 'lollipop' and 50 as 'CPC', with most misclassifications as 'irregular'. This problem is likely due to the model's handling of sparse data and its tendency to fit 'irregular' into these less common categories, since 'irregular' learned the most complex features since any unrecognizable shape is 'irregular'.

For the microphysical processes category (Fig. 9), IceDetectNet achieved an overall accuracy of 73 % (with a 9 % drop compared to the training dataset). The model still performed well in identifying 'pristine' ice crystals (82 %). In contrast, it shows a better performance in predicting 'aggregate' (17 % higher than in the training dataset) and 'aged-aggregate' (7 % higher than in the training dataset) ice crystals. This could be due to the changes in the data distribution, especially the changes in the aggregate fraction from 12 % in the training dataset to 37.7 % in the generalization dataset, which further emphasizes the importance of the balance of the dataset. After checking the main source of misprediction, we can see that underdetection still plays an important role, for example, the main source of misprediction of 'aggregate' is 'pristine', which is a typical misprediction problem.

## 5 Discussion

There are still opportunities to improve the generalization ability of IceDetectNet, particularly in the area of accurate detection and classification of aggregated ice crystals. In the case of imbalanced datasets, balancing strategies such as oversampling or undersampling techniques can be considered. Enriching the training dataset with a more comprehensive and diverse collection

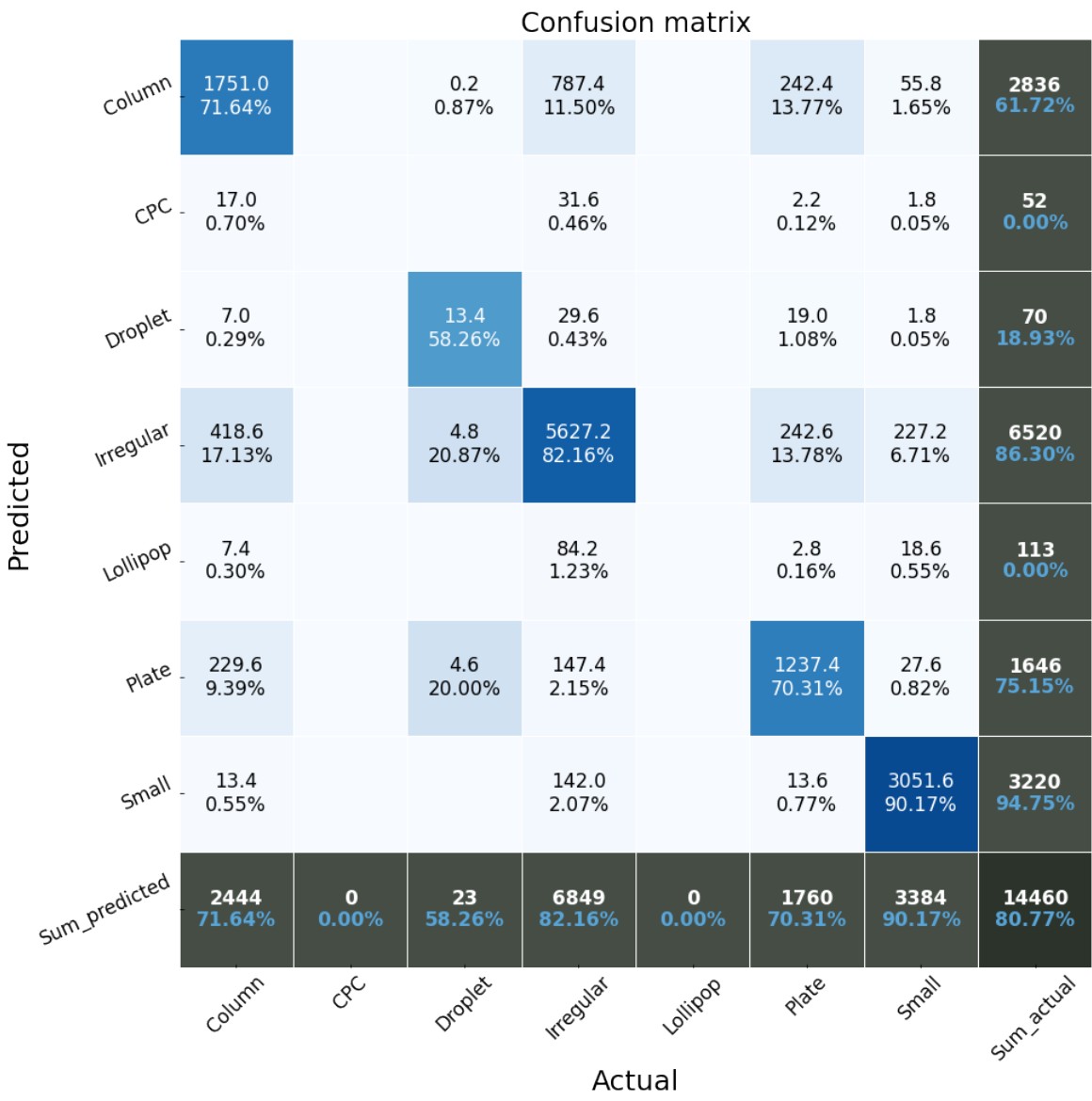

**Figure 8.** Similar confusion matrix as Fig. 4, but for generalization dataset. Class 'CPC' and 'Lollipop' are missing in the dataset and thus are 0 in its corresponding column.

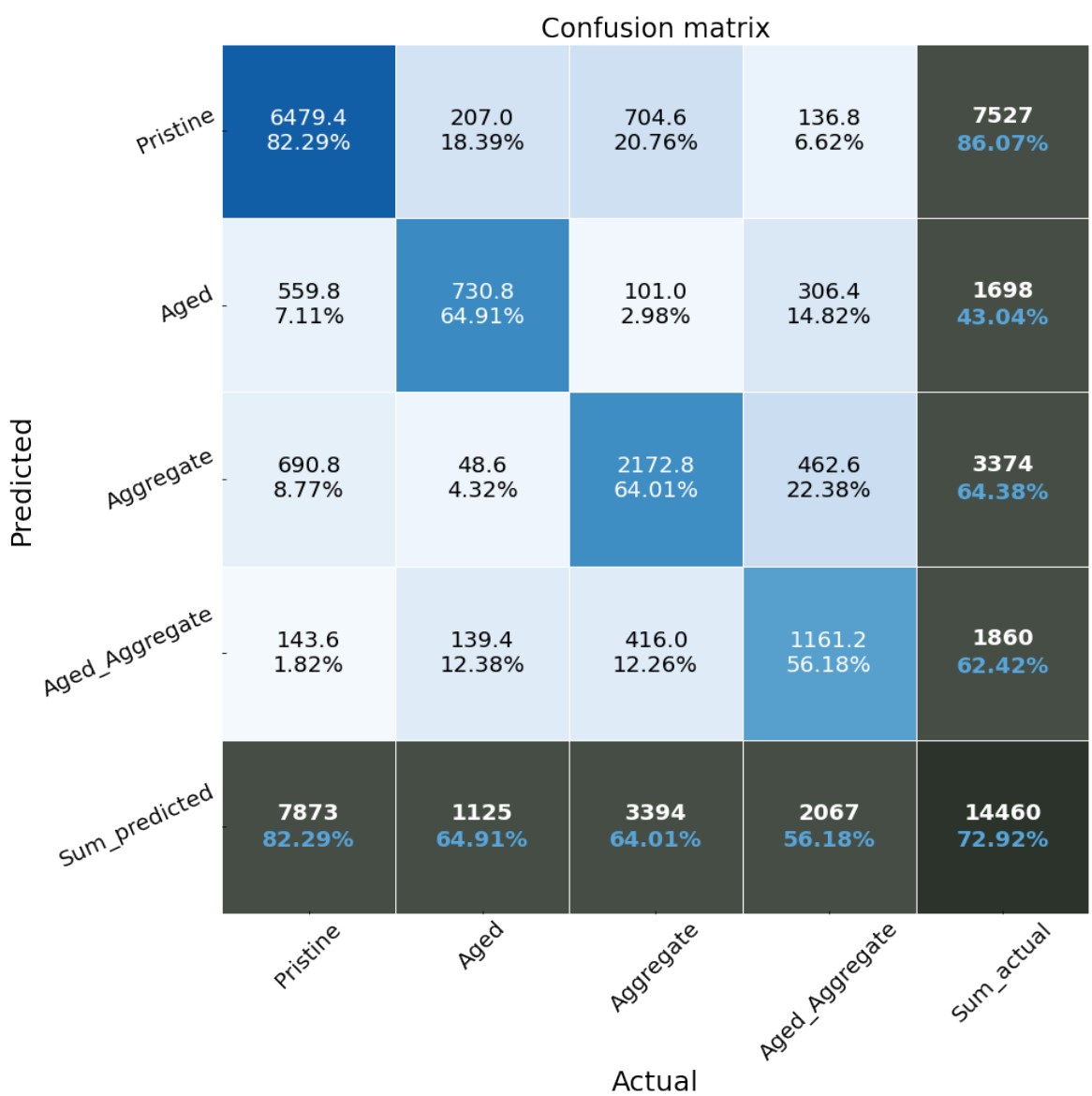

**Figure 9.** Similar confusion matrix as Fig. 5, but for generalization dataset.

of ice crystal data would improve the robustness and generalization capabilities of the algorithm. However, gathering more training data is a time-consuming process, and thus reducing the time needed for manual labeling of bounding boxes is an important task for future research. New techniques, including contrastive learning (Le-Khac et al., 2020) and unsupervised learning algorithms, should be investigated to reduce the need for extensive manual labeling. Furthermore, the efficacy of fine-tuning as an approach to include new ice crystal classes holds considerable promise for IceDetectNet. Fine-tuning - the process of adapting a pre-trained model (i.e. IceDetectNet) to a new, but still related, dataset - has been validated in a variety of fields as a means of achieving improved performance with comparatively smaller datasets (Tajbakhsh et al., 2016). This technique is consistent with the broader concept of transfer learning (Pan and Yang, 2009), which has seen widespread success in adapting models to new domains of application.

As the current dataset used for the training of IceDetectNet does not include some basic habits such as needles and rosettes, we plan to adapt IceDetectNet to include these categories once additional datasets containing these habits are available.

## 6 Conclusion

In this study we introduced IceDetectNet, a novel rotated object detection algorithm that is able to classify ice crystals not only on an image scale but down to the aggregate component scale. The algorithm was used in combination with a multi-label classification scheme, which assigns both a basic habit and microphysical processes to each ice component. The algorithm was trained and tested on two independent holographic ice crystal datasets, which were collected during the NASCENT campaign in Ny-Alesund, Svalbard.

The performance of IceDetectNet was evaluated in terms of its detection and classification performance, both on the image and aggregate component scale. At the image scale, IceDetectNet showed a good detection performance, correctly classifying 92 % of the ice crystals into the aggregate and non-aggregate classes. In terms of classification performance, it achieved an overall accuracy of 86 % for basic habits and 81 % for microphysical processes. Moreover, IceDetectNet achieved comparable classification accuracies as traditional deep learning algorithms on the non-aggregate subset, while the classification accuracies were lower for the aggregate subset. On the component scale, IceDetectNet showed good detection and classification performance across all bounding box sizes, indicating its ability to accurately classify components of aggregated ice crystals down to 662 pixels.

The generalization ability of IceDetectNet was examined on an independent generalization dataset that was collected during a different season. IceDetectNet showed good detection performance with an overall accuracy of 84 %. Although the classification accuracy decreased compared to the training dataset, the overall accuracies remained satisfactory for basic habits (81 %) and microphysical processes (72 %) classification. The aggregate subset showed lower performance compared to the non-aggregate subset, possibly due to imbalances in the dataset. This highlights the potential to further optimize the generalization ability of IceDetectNet through dataset balancing techniques, enlargement of the training dataset or fine-tuning.

However, the ice categories used in this study are specific to environmental and microphysical conditions present during the collection of the training data. In addition, the distinction between 'small' and 'irregular' ice categories combines both size and

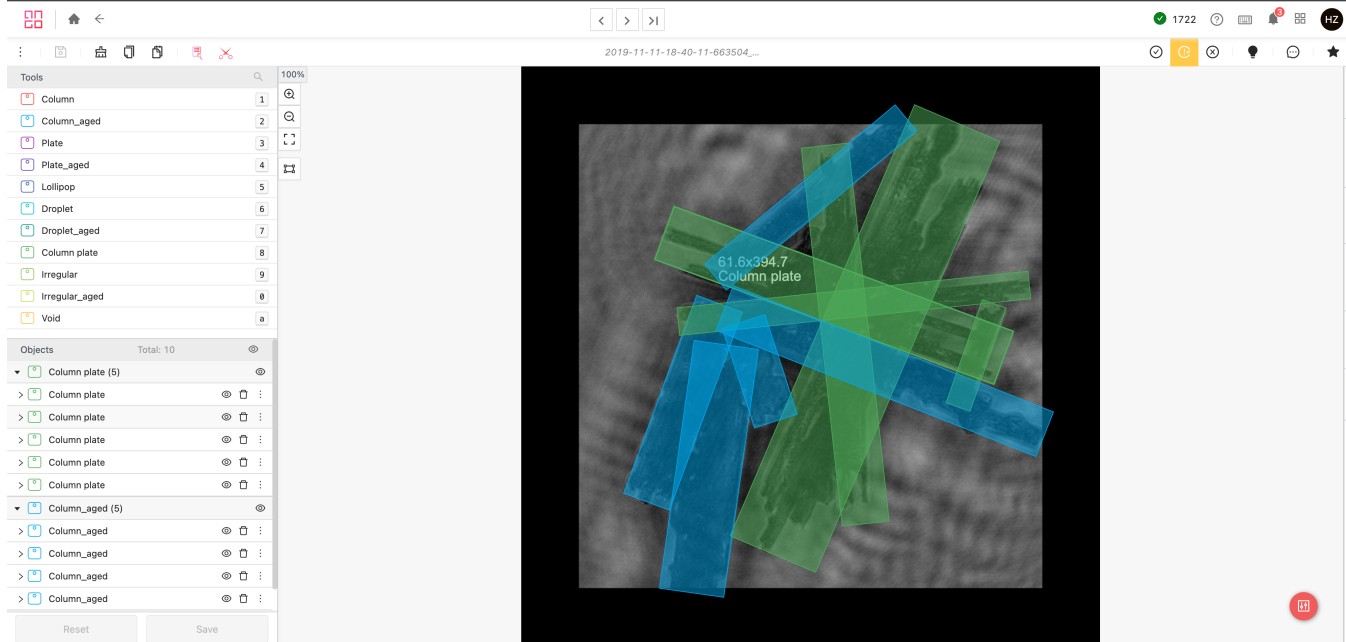

**Figure A1.** Overview of the graphical user interface for hand-labeling on hub.ango.ai's platform. Users can draw and adjust bounding boxes around components of aggregated ice and assign labels.

shape information, making it difficult to be classified. While these categories are appropriate for the current dataset, they may pose challenges when applying IceDetectNet to other datasets or comparing results with existing studies. However, adding or refining categories can be easily achieved through model fine-tuning.

IceDetectNet provides detailed shape information of the basic habit and microphysical processes down to the aggregate component scale of ice crystals and thus has the potential to improve the estimates of microphysical properties such as riming rate, aggreagtion rate and ice water content. Due to the good generalization ability of IceDetectNet, we expect that IceDetectNet can also be applied to other cloud imaging probes in connection with fine-tuning. This will help to better understand the radiative properties of clouds and the microphysical processes leading to precipitation formation.

*Code and data availability.* The code and data of this study are available upon request.

## Appendix A: Hand-labeling platform

An essential component of training the rotated object detection algorithm is the hand-labeling of bounding boxes and ice crystals, which was done through the hand-labeling platform created by (AngoAI, 2022). The platform offers a graphical user interface to draw bounding boxes, adjust their size and rotation, and assign labels (Fig. A1).

## Appendix B: Detailed criteria for ice crystal classification

The classification of ice crystals into their respective basic habits and microphysical processes is a challenging task that requires a set of rules to ensure consistency and accuracy across the dataset. Here we describe the criteria used for the multi-label classification of ice crystals. We randomly select several images from each category as examples (see Fig. B1) and present the process of how we hand-label an ice crystal (see Fig. B2).

The classification process begins by using human judgment to determine whether the ice particle is an aggregate that contains
515 more than one component. If an ice crystal is not aggregated, the classification process proceeds directly with the classification of the basic habit. For aggregated crystals, the process differs between training and evaluation of IceDetectNet. In training, each component is manually located with a bounding box (i.e. smallest rectangle box) around the component and these boxes are then classified. In the multi-label classification, only the largest visually identified component of the aggregate is classified, without drawing a bounding box. The classified basic habit of this largest component will represent the basic habit of the whole
aggregate ice crystal.

The next step is to classify the basic habits of the ice crystals/components. If the basic habit is not recognizable (as defined in Table,1), the size of the ice/component is assessed by eye. Small crystals are classified as 'small' and all others as 'irregular-aged'. If the basic habit is recognizable, we classify based on shape. Special shapes, like 'lollipop-aged' for lollipop-like crystals or 'frozen droplets' for those with droplet features, are classified first. Rectangular-shaped ice crystals (with 4 distinct
edges) are classified as 'columns', whereas rectangular-shaped ice crystals with multiple branches at the end of the maximum dimension are labeled 'CPC-aged. Note, that the CPC-aged categories also include needle bundles with missing plate sections. Hexagonal crystals (with 6 distinct edges) are classified based on their aspect ratio, where a high aspect ratio indicates a 'column,' and a low aspect ratio a 'plate' Crystals/components that don't fit these categories are considered 'irregular-aged'.

Once the basic habit is determined, the appearance of the edges of the ice/component determines whether the ice/component
is aged or not. As mentioned earlier (see section. 2), 'irregular', 'CPC', and 'lollipop-aged' are aged by default, while small is 'pristine' by definition. So we only need to decide if 'column', 'plate' and 'frozen drops' are aged or not. Usually, when an ice/component is aged, it has some tiny bumps on the edges.

*Author contributions.* HZ and XL collaborated closely to design and develop the algorithm architecture, and they conducted all the runs. FR and JH actively participated in discussions and gave suggestions regarding the development of the algorithm. JP collected the training
and test data with the assistance of ROD and JH and conducted the single-label hand-labeling on the image scale. Both HZ and XL labeled the bounding boxes for the training and test data. Subsequently, HZ re-labeled the train and test data based on the proposed multi-label classification scheme. The manuscript was written by HZ, with valuable input and discussions from FR, JH, ROD and XL. The initial idea for the classification scheme was conceived through discussions between ROD and HZ.

*Competing interests.* The authors declare that they have no conflict of interest.

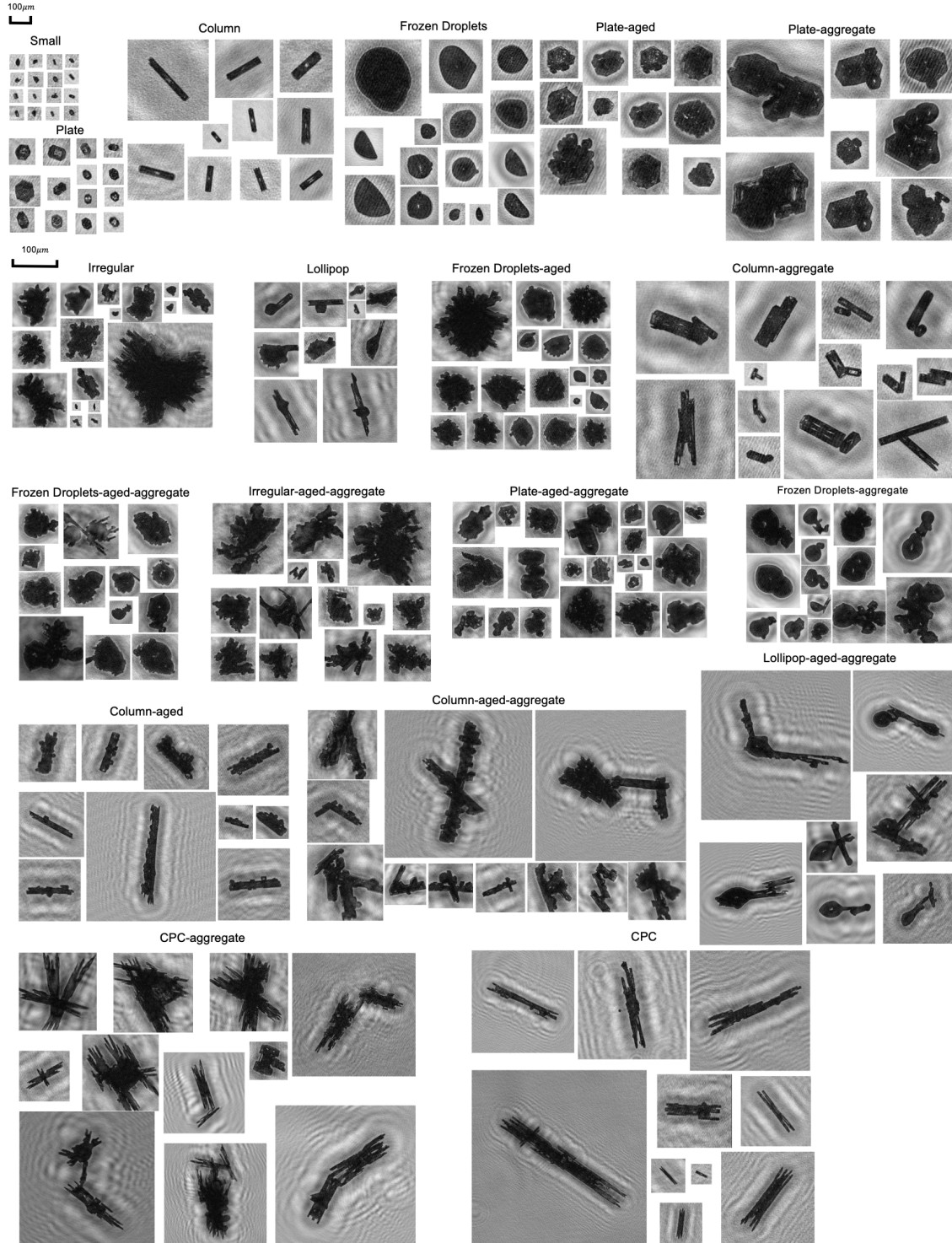

**Figure B1.** A randomly selected sample of ice crystal images from each category based on the multi-label classification scheme

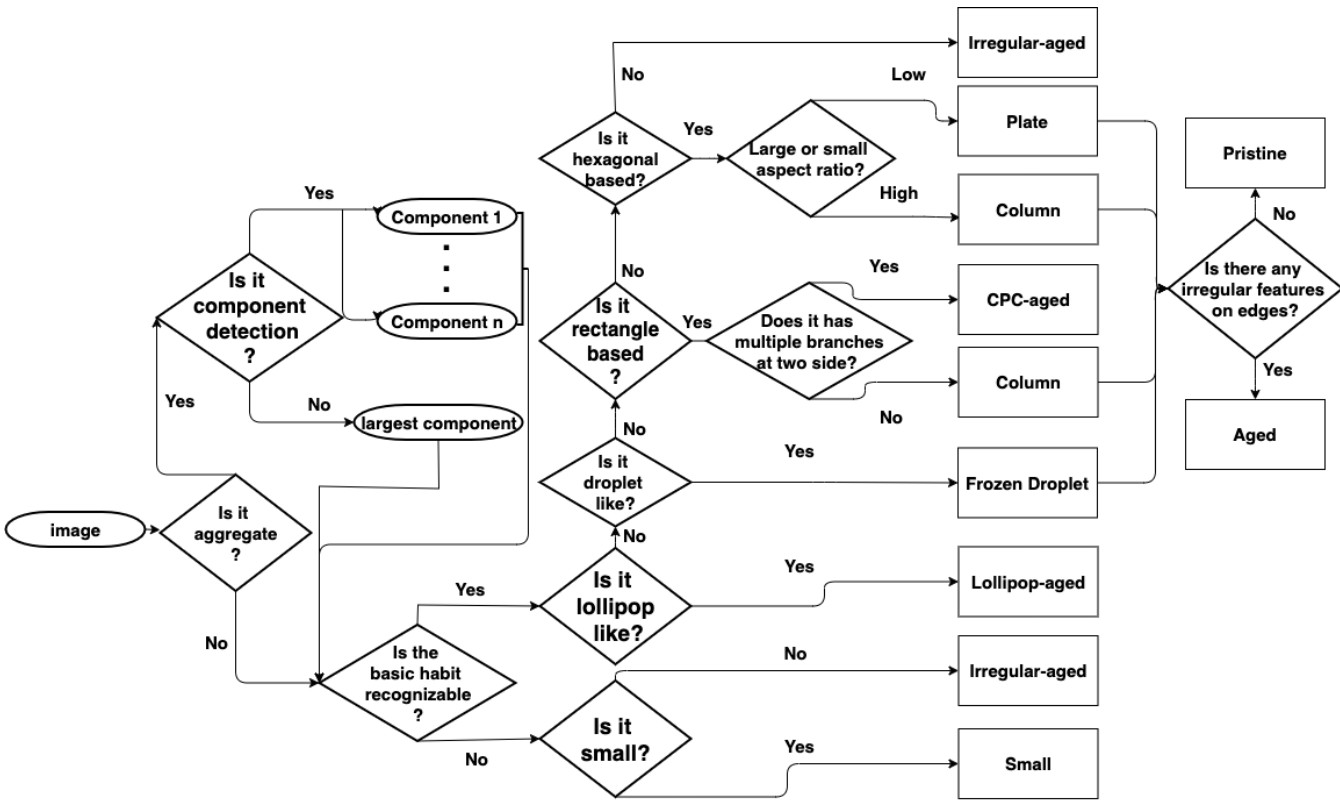

**Figure B2.** The process of classifying ice crystals

*Disclaimer.* TEXT

*Acknowledgements.* Firstly, the authors extend their heartfelt gratitude to Trude Storelvmo, and Alexander Binder for their invaluable guidance during HZ's Master's study, which is the foundation of this study. Also, we acknowledge the financial support from the European Research Council (ERC) under the European Union's Horizon 2020 research and innovation program (grant No. 101021272). ROD would like to acknowledge EEARO-NO-2019-0423/IceSafari, contract no. 31/2020 of EEA Grants/Norway Grants and EUs HORIZON-WIDERA 2021 program with project number 101079385 for financial support.

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
