# Peer review of "IceDetectNet: A rotated object detection algorithm for classifying components of aggregated ice crystals with a multi-label classification scheme"

_EGUsphere, 2023_

## Author Comment (AC1)

**Authors' comments to Anonymous Referee #2**

We would like to thank the peer reviewer for the thorough review of our manuscript and the insightful feedback. These comments have significantly improved the quality of our work. In the following sections, we present the reviewer's comments (in black), our responses (in red), and the changes made in the revised manuscript (in blue). Please note that all line numbers in our responses correspond to those in the revised manuscript.
* * *
**Overall comments:**

The manuscript "IceDetectNet: A rotated object detection algorithm for classifying components of aggregated ice crystals with a multi-label classification scheme" by Zhang et al. presents a new deep learning algorithm that classifies each component of aggregated ice crystals based on their basic habit and physical processes. This algorithm enables a more detailed classification of the ice crystals than conventional algorithms and thus provides an innovative and improved tool for identifying atmospheric ice crystals. The algorithm and its evaluation are well described, individual steps are listed in detail, which makes the content of the manuscript very comprehensible and easy to follow. I recommend this paper for publication after major revision.

1. - L. 75: "Following this initial categorization, each ice crystal was classified into one of seven basic habits: 'column', 'plate', 'lollipop' (Pasquier et al., 2022a), 'Columns on Capped-Columns' (CPC, Pasquier et al. 2023), 'irregular', 'frozen droplets', and 'small'." Can the authors explain and justify why this ice particle classification is used? What about other ice crystals occurring in the atmosphere, e.g., needles, droxtals, rosettes, …? Why are the ice particles classified in specific (e.g. lollipop) and unspecific shapes (e.g. irregular)? This shape categorization results in a strong unbalanced training data set. I see, the authors consider this imbalance in their analyses. However, a better classification of particles will counteract the imbalance and might increase the classification performance of IceDetectNet and its applicability to other independent test datasets.

   We agree with the reviewer that a clearer justification for the classes used in this study needs to be provided. Therefore, and in line with the same concern raised by reviewer 1, we have now clearly stated in the main text:

**L83-85:**

"Our seven basic habit categories were determined by their presence and distinct shape features observed in our dataset collected in Arctic mixed-phase clouds in NyAlesund. These basic habit classes are based on the categories used in Pasquier et al., 2022b as we used the same dataset."

While we acknowledge that the 'CPC' and 'lollipop' categories may appear to be more specialized, their inclusion was intentional and aimed at understanding the microphysical processes responsible for their creation (Pasquier et al. 2023). This allows IceDetectNet to maintain high performance on well-defined categories, while still providing a mechanism to classify ice crystals that do not fit nicely into these commonly used shapes.

We understand the critical role that a wider range of ice crystal habits, such as needles, play in mixed-phase clouds. The absence of these habits in our dataset limits the inclusion of these habits in the classification scheme and in the training and validation process of IceDetectNet. In our extended discussion (see below), we highlight the potential of fine-tuning IceDetectNet to integrate ice crystal habits that were not part of the original training set.

**L94-99:**

"For a more detailed discussion of the categorization criteria, we refer to Appendix B. However, due to data limitations, our dataset does not capture every possible basic ice habit, such as needles and rosettes. This limitation is acknowledged and further discussed in Sect.5 where we look at potential extensions to IceDetectNet. As new data containing additional ice habits become available, IceDetectNet can be updated as it is designed to incorporate these new habits through fine-tuning, ensuring the continued evolution of the model."

**L462-468:**

"As the current dataset used for the training of IceDetectNet does not include some basic habits such as needles and rosettes, we plan to adapt IceDetectNet to include these categories once additional datasets containing these habits are available. "

Regarding the applicability of IceDetectNet, on the validation subset, IceDetectNet achieved 92% detection accuracy and classified ice crystals with 86% accuracy for basic habits and 81% accuracy for microphysical

processes. Also, it performed comparably to traditional algorithms for non-aggregates. These could state that IceDetectNet has good performance.

2. - While the evaluation of the model performance based on the training data set is extensive (Sect. 4.1-4.2), the application of the IceDetectNet algorithm to the test data set is far too short. A more detailed analysis of the generalization ability of IceDetectNet would be desirable. How accurate is the basic habit classification and physical process classification for the test data set? Why do the individual values of evaluation parameters change when the test data set is used instead of the training data set? What are the reasons? Where are (no) challenges, issues? What conclusions can the authors draw from the evaluation of IceDetectNet using the independent test dataset? How well could the algorithm be applied to other test data sets from different seasons, locations, … ? Do the authors expect any limitations here?

Thanks for pointing this out and we agree that we did not discuss the test data set thoroughly. Also, test dataset sounds a bit confusing. With this in mind, we have renamed the test dataset to generalization dataset, explicitly described the division of the training set and updated the main text accordingly.

To clarify, here we use two different datasets: a training dataset, and a generalization dataset, each of which serves different purposes.

1. Training and validation subsets: This subset allows us to perform cross-validation and ensure that the model performs consistently within the known parameters of the dataset.

2. Generalization dataset: The generalization dataset is separate from the training process. It is designed to evaluate the model's ability to generalize across different environmental conditions, especially those not represented in the training dataset. This dataset was intentionally collected under significantly different conditions (from -23 to -15°C) to challenge the model with ice particle types that may not be present in the training data. We acknowledge that this dataset does not include classes such as 'lollipop' or 'CPC' due to their absence under the specific conditions and seasons of collection.

**L100-106:**

"The dataset collected on November 11, 2019 (Pasquier et al., 2022b), hereafter training dataset,  was used to train IceDetectNet. During the training it was divided into a training subset (comprising 80% of the data) and a validation subset (made up of the remaining 20%), using a cross-validation

method (detailed introduction in Sect. 3.8.1). This validation subset serves a similar purpose as the traditional test sets used in other studies (Jaffeux et al., 2022; Xiao et al., 2019; Touloupas et al., 2020), providing an initial evaluation of the model's performance under known conditions. On the other hand, the generalization dataset was collected on a different date, April 1, 2020 (Pasquier et al., 2022b) which is not used during training but to evaluate the generalization abilities of IceDetectNet."

Moreover, we have now added the confusion matrix plots for the generalization dataset and added the analysis on the plots as follows:

**L433-456:**

"To gain further insights into IceDetectNet's performance in each ice category, here we analyzed the confusion matrices (mean of 5 models) for basic habit classification (Fig.9) and microphysical processes classification (Fig.10) for the generalization dataset as well.

IceDetectNet achieved an overall accuracy of 81% for the basic habit categories (Fig.9) The confusion matrix shows that IceDetectNet still performed better for the ice categories that comprise a large fraction of the dataset, like 'small' (precision of 90%). However, among these, 'column' classification performance had a large performance drop (18% decrease in precision compared to the training dataset), and its two main misprediction sources were 'irregular' and 'plate', which almost represent all the mispredictions (29%). This could be due to the data distribution shift from 'column' to 'plate'. Under the general decrease trend among all categories, 'irregular' surprisingly has a 10% increase in precision. The main misprediction of 'irregular' comes from 'column' in both the training dataset and generalization data, which could be the reason that IceDetectNet learned many column features in the training dataset and thus distributed higher weights on these column features. While the number of 'column' is much less in the generalization dataset and thus leads to better performance in classifying 'irregular'. For the missing categories like 'lollipop' and 'CPC' that had zero actual occurrences, IceDetectNet still predicted 113 ice as 'lollipop' and 50 as 'CPC', with most misclassifications as 'irregular'. This problem is likely due to the model's handling of sparse data and its tendency to fit 'irregular' into these less common categories, since 'irregular' learned the most complex features since any unrecognizable shape is 'irregular'.

For the microphysical processes category (Fig.10), IceDetectNet achieved an overall accuracy of 73% (with a 9% drop compared to the training dataset). The model still performed well in identifying 'pristine' ice crystals (82%). In contrast, it shows a better performance in predicting 'aggregate' (17% higher than in the training dataset) and 'aged-aggregate' (7% higher than in the training dataset) ice crystals. This could be due to the changes in the data distribution, especially the changes in the aggregate fraction from 12% in the training dataset to 37.7% in the generalization dataset, which further emphasizes the importance of the balance of the dataset. After checking the main source of misprediction, we can see that underdetection still plays an important role, for example, the main source of misprediction of 'aggregate' is 'pristine', which is a typical misprediction problem."

[Figure]

*Figure 1 Similar confusion matrix as Fig. 5 in the manuscript, but for generalization dataset. Class 'CPC' and 'Lollipop' are missing in the dataset and thus are 0 in its corresponding column.*

[Figure]

*Figure 2 Similar confusion matrix as Fig. 6 in the manuscript, but for generalization dataset*

3. - L. 38: "generalization abilities": This term is too broad to be understandable. Although this term will be explained later it would be useful to use a more specific formulation here.

Thanks for pointing this out. We have revised the sentence to make "generalization abilities" clearer. Specifically, we have added a description that emphasizes the dependence of an algorithm's performance on the characteristics of the training dataset and the subsequent need for threshold adjustments when encountering new datasets.

**L41-45:**

"Furthermore, these algorithms demonstrated limitations in their ability to perform effectively on different datasets, as their classification performance was strongly influenced by the characteristics of the training dataset (Bishop and Nasrabadi, 2006; Goodfellow et al., 2016), which is defined as the generalization ability of the models. This dependency requires significant adjustments to the optimal thresholds when these algorithms are applied to new, unseen datasets."

4. - L.34: Square area of the image, particle maximum dimension, and area ratio do not classify the shape of ice crystals, they define the size of ice crystals. Please correct.

Thanks for catching this, we have now rephrased this to:

**L36-38:**
"Early ice crystal classification techniques used simple features like edge complexity (Cunningham, 1978), circular deficiency (Rahman et al., 1981), the surface area and perimeter (Duroure et al., 1994), the complexity (combined several geometric features such as particle area and area ratio (Schmitt and Heymsfield, 2014) to classify the shape of ice crystals,"

5. - L. 85: Please provide more information about the training and test data set (location, meteorological conditions, ...). How representative are both data sets for the occurrence of generally possible ice particles in the atmosphere (in all seasons and locations)?

The detailed location information and meteorological conditions are included in Pasquier et al., 2022b. We also briefly introduced the information in:

**Line 118-122:**

"The difference between the training and generalization datasets is an example of the natural variability of field observations. In our case, the two datasets were collected during different seasons, resulting in variations in the environmental conditions. The training dataset was collected in the temperature range from -8 to -3 ℃ (mostly in the column regime) while the generalization dataset was collected between -23 and -15℃ (mostly in the plate regime) (Pasquier et al., 2022b) These differences allow us to assess the generalization ability of IceDetectNet and to examine its performance in diverse environmental conditions."

Regarding the representativeness of the data, our seven basic habit categories were determined by their presence and distinct shape features

observed in our dataset collected in Arctic mixed-phase clouds in NyAlesund, which is representative of these measurements conducted in NyAlesund (Pasquier et al., 2022b) but do not capture the full range of conditions. Also, to clarify, the main goal of this paper is to introduce a novel technique (IceDetectNet) that could classify ice crystal components.

6. - L. 156: "Every bounding box was visually classified in an ice category following the multi-label classification scheme". In Fig. 1, some ice particles of different categories look similar. How does a visual, i.e., subjective, classification scheme influence the training of the data set? How is it decided that an ice particle that could apparently fit into two classes is assigned to one class?

Thank you for your question about our classification method. In our study, visual classification was used as a first step to categorize ice particles based on observable features. We realized that this method involves a degree of subjectivity that could potentially affect the consistency of the training dataset and, consequently, the performance of the model. Thus, we added detailed classification guidelines to ensure consistency across categorizations. These guidelines include specific visual indicators for each category. Details are provided in '**Appendix B: Detailed criteria for ice crystal classification**' which explains the criteria for defining each category. Also, we have conducted several rounds of review where misclassified ice particles are classified and relabeled to ensure accuracy and consistency. However, we acknowledge that people can never be perfect but that the training dataset will improve as more data is incorporated. Comparing the difference between the first round of hand-labeling and the label after reviews, we estimate the misclassification to be around 5%.

**L502-523:**

**Appendix B: Detailed criteria for ice crystal classification**

The classification of ice crystals into their respective basic habits and microphysical processes is a challenging task that requires a set of rules to ensure consistency and accuracy across the dataset. Here we describe the criteria used for the multi-label classification of ice crystals. We randomly select several images from each category as examples (see Fig. B2) and present the process of how we hand-label an ice crystal (see Fig. B1).

The classification process begins by using human judgment to determine whether the ice particle is an aggregate that contains more than one component. If an ice crystal is not aggregated, the classification process proceeds directly with the classification of the basic habit. For aggregated crystals, the process differs between training and evaluation of IceDetectNet. In training, each component is manually located with a bounding box (i.e. smallest rectangle box) around the component and these boxes are then classified. In multi-label classification, however, only the largest visually identified component of the aggregate will be classified without drawing a bounding box.

The next step is to classify the basic habits of the ice crystals/components. If the basic habit is not recognizable (as defined in Table,1), the size of the ice/component is assessed by eye. Small crystals are classified as 'small' and all others as 'irregular-aged'. If the basic habit is recognizable, we classify based on shape. Special shapes, like 'lollipop-aged' for lollipop-like crystals or 'frozen droplets' for those with droplet features, are classified first. Rectangular shapes with multiple branches at two sides (maximum dimension side) are labeled 'CPC-aged,' and others as 'column.' Hexagonal crystals are classified by aspect ratio: normally, a high ratio indicates a 'column,' and if a lower one with clear hexagonal patterns is a 'plate', otherwise, a 'column' again. Crystals/components that don't fit these categories are considered 'irregular-aged'.

Once the basic habit is determined, the appearance of the edges of the ice/component determines whether the ice/component is aged or not. As mentioned earlier (see section. 2) 'irregular', 'CPC', and 'lollipop-aged' are aged by default, while small is 'pristine' by definition. So we only need to decide if 'column', 'plate' and 'frozen drops' are aged or not. Usually, when an ice/component is aged, it has some tiny bumps on the edges.

[Figure]

*Figure 3 A randomly selected sample of ice crystal images from each category*

[Figure]

*Figure 4 The process of classifying ice crystals*

7. - L. 162: "The initial image needs to be enlarged by 15 %". What is the reference? Area? Length, width?

   Thank you for your comment about the enlargement of the original image. To clarify, the enlargement applies to both the length and width of the image, each increased by 15%. We refined the original sentence as follows:

   **L182-184:**

   "Before the image is fed into IceDetectNet, the initial image is enlarged by 15% (in both length and width) to ensure full coverage of bounding box drawing and maintain the aspect ratio (see input in Fig.3) by adding black pixels (pixel values = 1) around the borders."

8. - L. 165: "All images are then uniformly resized to 512x512 pixels." How this is done?

   Thank you for your question regarding the image resizing process described in our manuscript.

   After the initial cropping to a square format, the images are then resized to a uniform dimension of 512x512 pixels. This resizing is done using the PyTorch library's resize function, which uses bilinear interpolation. Bilinear interpolation is a widely used technique in image processing that calculates the output pixel

value in the resized image as a weighted average of the pixels in the nearest 2x2 neighborhood in the original image. This method is particularly effective for scaling images as it helps to preserve the smoothness and detail of the original image while minimizing potential artifacts such as pixelation or loss of fine detail. By uniformly resizing the images to 512x512 pixels, we ensure a consistent input size for the neural network, which is a standard practice in deep learning to maintain uniformity of the input layer and optimize computational efficiency. This consistency is also critical for batch processing during model training and inference.

In response to your question, we rephrased:

**Line 185-186:**
"To ensure consistency across the network training and testing, all images are then uniformly resized to 512 × 512 pixels by using bilinear interpolation after the enlargement."

9. - L. 168: "We replicate the single dimension three times to emulate the three-dimensional structure of RGB images". I understand why the authors are doing this. However, how do results change, if zero-arrays are used in the second and third dimension? Won't a triple replication as the authors do lead to a loss of contrast in the resulting RGB image?

This is a good point, in our methodology, the replication of the single dimension across the three channels of an RGB image is designed to fully utilize pre-trained models that inherently expect RGB input. Using zero arrays for the second and third dimensions, as opposed to replication, would significantly alter the structure of the input. While this approach preserves the original data in one channel, it introduces a complete lack of information in the other two, which could lead to suboptimal use of the capabilities of the pre-trained network, as these models are tuned to detect and interpret correlations and patterns across all three channels.

Regarding the concern about loss of contrast, it's important to distinguish between the perceptual contrast in the visual representation and the contrast relevant to a model's learning process. Channel replication maintains the same relative contrast between pixels as in the original single-channel image. Although this may appear to result in a loss of contrast in the conventional sense (since all channels are identical, resulting in a grey-scale image when viewed), for the model the informational contrast between different regions of the image remains intact and actionable.

10. - Fig. 3: Why do duplicate bounding boxes have to be removed in step 6, when the duplicate bounding boxes should have already been removed in step 3?

Good catch, we need to remove the duplicate bounding boxes in two steps to ensure accuracy and reduce redundancy throughout the object detection process at the same time. Specifically, in step 3, we address potential duplicate bounding boxes that arise during the training process. At each ice component position, multiple bounding boxes will be predicted. During this phase, the predicted bounding boxes are matched to the hand-labeled bounding boxes based on their Intersection over Union (IoU) values. If the IoU of a bounding box with the hand-labeled bounding box exceeds a predefined threshold (50%), it is considered a match, and the training target is adjusted to match the hand-labeled bounding box. This step may result in multiple bounding boxes at a single location matching one hand-labeled bounding box, resulting in multiple bounding boxes exceeding the IoU threshold.

A further refinement step is required in step 6. At this point, we perform a comparison among all predicted bounding boxes instead of with hand-labelled bounding boxes. If two predicted bounding boxes have an IoU over 50%, then only the precited bounding box with the higher confidence level of classification will be kept. This additional filtering step is critical to improving the accuracy of our object detection algorithm, ensuring that each detected object is represented by a single, most likely bounding box.

By implementing this two-step approach, we significantly reduce the likelihood of duplicate detections, thereby improving the accuracy and reliability of our model's predictions. We appreciate your attention to this detail and hope this explanation and the following rephrase clarifies the logic behind our methodology.

Rephrased:

**L1201-204:**

"After classification, a post-processing step is performed to further remove duplicate bounding boxes (Fig.3 step 6) by comparing all predicted bounding boxes instead of hand-labeled bounding boxes using the Intersection over Union (IoU) threshold and the confidence level of classification."

11. - L. 208-213: Can the authors briefly explain why the learning rate and epochs are chosen in this way?

We have chosen to start the training with a learning rate of 0, increasing linearly to 0.0025 over the first 500 steps, which is a technique (Gotmare et al. 2018) known as learning rate warm-up. This approach helps to stabilize the training of the model early on and reduces the risk of divergent behavior by gradually scaling the parameter updates. This method is particularly useful for complex models such as IceDetectNet or datasets where abrupt, significant adjustments early in training can lead to instability.

At the 500-step mark, the learning rate stabilizes at 0.0025. This steady phase allows the model to converge to a potentially good solution space with consistent update sizes. The choice of 0.0025 is a balance between learning rates that are too slow (where the model would take an excessive number of iterations to converge) and too fast (where the model might overshoot optimal solutions).

The subsequent reduction of the learning rate at predetermined epochs is a strategy known as learning rate annealing or decay. This method is crucial for fine-tuning the model. As training progresses and the model approaches optimal performance, smaller updates become necessary to refine and fine-tune the model parameters, preventing overshooting and facilitating convergence to a more accurate solution. The specific epochs for reduction - 64th and 88th - are chosen based on empirical evidence and the particular training dynamics of IceDetectNet, where these points mark phases of training where further reduction of the learning rate helps to stabilize and refine the learning process.

In summary, the structured progression from a warm-up phase to a constant learning rate, followed by strategic reductions, is designed to lead to optimal performance while avoiding common pitfalls such as getting trapped in local minima or experiencing erratic parameter updates. These decisions (i.e. learning rate, epoch) are often empirical, influenced by the model's response to training, and aim to achieve a fine balance that promotes effective learning throughout the training process.

12. - Equation 2: What are correctly predicted positive and negative instances when ice crystal classes are predicted?

Thanks for catching this. We agree that this was confusing so have rephrased this to remove positive and negative instances completely and made the whole paper consistent. See the updated evaluation metrics subsubsection:

**L259-282:**

**3.8.2 Evaluation Metrics**

To assess the performance of IceDetectNet, we employ several metrics that evaluate the model performance with regard to different aspects, including overall accuracy, precision, recall, confusion matrix and F1 score. The overall accuracy is defined as the ratio of the number of correct predictions to the total number of particles (Goodfellow et al., 2016). An overall accuracy of 100% means that, for example, all ice particles were correctly predicted, while an overall accuracy of 0% indicates that all particles were mispredicted. While overall accuracy provides a quick and straightforward metric to interpret the model performance, it can be misleading when dealing with imbalanced datasets where classes are not equally represented. In such cases, the model may perform well in predicting the dominant classes but struggle with predicting rare classes. Precision and recall both measure the accuracy of a deep-learning classification model in predicting a single category from two perspectives. Precision is calculated as the ratio of the number of correct predictions of a specific class to the total number of predictions (Goodfellow et al., 2016), while recall is defined by the ratio of the number of correct predictions of a specific class to the total number of this class (Goodfellow et al., 2016). A high precision score indicates effective identification of a specific class, while a high recall score indicates that the model excels in identifying instances of a particular class and is less likely to miss relevant instances that belong to the class. All of these metrics can be combined and visualized in a so-called confusion matrix (Goodfellow et al., 2016). In a confusion matrix, the diagonal, from top-left to bottom-right, corresponds to correct predictions made by the model, while the elements outside this diagonal represent misclassifications. The bottom-right cell of the matrix displays the total number of ice crystals and the overall accuracy. The bottom row provides the actual counts per class and their respective per-class precision. Similarly, the rightmost column presents the predicted counts per class and the associated per-class recall. The F1 score is a harmonized metric that combines precision and recall, providing a balanced measure of a model's performance, particularly in situations where the balance between precision and recall is critical (Goodfellow et al., 2016). This score reaches its best value at 1 (indicating perfect precision and recall) and its worst value at 0. In the context of IceDetectNet, a high F1 score would indicate not only that the model accurately identifies ice particles (high precision), but also that it successfully detects the majority of actual ice particles (high recall), making it a robust

metric for evaluating model performance across different classes, especially in the presence of imbalanced datasets.

Reference:

Goodfellow I, Bengio Y, Courville A. Deep learning[M]. MIT press, 2016.

Gotmare A, Keskar N S, Xiong C, et al. A closer look at deep learning heuristics: Learning rate restarts, warmup and distillation[J]. arXiv preprint arXiv:1810.13243, 2018.

Pasquier, J. T., Henneberger, J., Ramelli, F., Lauber, A., David, R. O., Wieder, J., Carlsen, T., Gierens, R., Maturilli, M., and Lohmann, U.:
Conditions favorable for secondary ice production in Arctic mixed-phase clouds, Atmospheric Chemistry and Physics, 22, 15 579–15 601,2022b.

Pasquier, J. T., Henneberger, J., Korolev, A., Ramelli, F., Wieder, J., Lauber, A., Li, G., David, R. O., Carlsen, T., Gierens, R., et al.:
Understanding the history of two complex ice crystal habits deduced from a holographic imager, Geophysical Research Letters, 50,
e2022GL100 247, 2023

---

## Author Comment (AC2)

**Authors' comments to Anonymous Referee #1**

We would like to thank the reviewer for the thorough review of our manuscript and insightful feedback. These comments have significantly improved the quality of our work. In the following sections, we present the reviewer's comments (in black), our responses (in red), and the changes made in the revised manuscript (in blue). Please note that all line numbers in our responses correspond to those in the revised manuscript.

**Overall comments:**

The authors present the IceDetectNet, a new deep learning algorithm aimed at refining the classification of ice crystals, which is significant for understanding cloud properties and precipitation processes. The need for improved classification methods arises from the challenges associated with current deep learning approaches, such as the difficulty in distinguishing individual components within aggregated ice crystals and the compromise between classifying basic habits and microphysical processes. IceDetectNet attempts to address these issues by integrating a rotated object detection technique for component-specific analysis and a multi-label classification framework. The approach is innovative and classification of individual components of aggregated crystals and the related microphysical processes is an important upgrade to traditional machine learning approaches. Yet, the study contains severe limitations, which affects its generalization ability. The main shortcoming is the choice of ice crystal habit and microphysical process categories, which include specific habit categories while omitting other frequently observed ice crystal habits. Since the habit classification scheme is the backbone of this study, this should be well thought of and justified in order for the algorithm to be generazible. It is recommended that the authors refine the the habit classification scheme and address other major concerns related to the structure of IceDetectNet before the paper can be recommended for publication.

1. The habit classes that the authors are proposing contains seven basic habits. Yet, the choice of these basic habit categories is not justified or based on classification schemes presented in the literature (e.g. Kikuchi et al., 2013; Hallett & Bailey, 2009). Some habit classes, like "CPC" or "lollipop" are rather specialized, whereas other key habit classes (especially for mixed-phase clouds), like needles, are missing. Since the manuscript is focused on identification of microphysical processes and crystal growth regimes, it would be justified to classify the crystals either as single crystals (typically encountered below -20°C; see Hallett & Bailey, 2009) and polycrystalline

habits (typically encountered in colder temperatures) and/or as plate-like (colder than -10°C or warmer than -2°C), column-like (warmer than -10°C) or mixed-habits. More generalized habit classification scheme would enhance the usability of the algorithm for other campaigns.

Thank you for your thoughtful comments and for pointing out the differences of our classification approach relative to other established classification schemes, such as Hallett & Bailey (2009).  The main goal of this paper is to introduce a novel technique (IceDetectNet) that can classify ice crystal components, which is not meant to be a completed/finished model including all ice habits.

Our seven basic habit categories were determined by their presence and distinct shape features observed in our dataset collected in Arctic mixed-phase clouds in NyAlesund. These basic habit classes are based on the categories used in Pasquier et al., 2022b as we used the same dataset. Following your advice, we have included an additional section in the appendix (shown below) to explain the classification criteria and rationale behind the definition of each ice category.

**L502-523:**

**Appendix B: Detailed criteria for ice crystal classification**
The classification of ice crystals into their respective basic habits and microphysical processes is a challenging task that requires a set of rules to ensure consistency and accuracy across the dataset. Here we describe the criteria used for the multi-label classification of ice crystals. We randomly select several images from each category as examples (see Fig. B2) and present the process of how we hand-label an ice crystal (see Fig. B1).

The classification process begins by using human judgment to determine whether the ice particle is an aggregate that contains more than one component. If an ice crystal is not aggregated, the classification process proceeds directly with the classification of the basic habit. For aggregated crystals, the process differs between training and evaluation of IceDetectNet. In training, each component is manually located with a bounding box (i.e. smallest rectangle box) around the component and these boxes are then classified. In multi-label classification, however, only the largest visually identified component of the aggregate will be classified without drawing a bounding box.

The next step is to classify the basic habits of the ice crystals/components. If the basic habit is not recognizable (as defined in Table,1), the size of the ice/component is assessed by eye. Small crystals are classified as 'small' and all others as 'irregular-aged'. If the basic habit is recognizable, we classify based on shape. Special shapes, like 'lollipop-aged' for lollipop-like crystals or 'frozen droplets' for those with droplet features, are classified first. Rectangular shapes with multiple branches at two sides (maximum dimension side) are labeled 'CPC-aged,' and others as 'column.' Hexagonal crystals are classified by aspect ratio: normally, a high ratio indicates a 'column,' and if a lower one with clear hexagonal patterns is a 'plate', otherwise, a 'column' again. Crystals/components that don't fit these categories are considered 'irregular-aged'.

Once the basic habit is determined, the appearance of the edges of the ice/component determines whether the ice/component is aged or not. As mentioned earlier (see section. 2) 'irregular', 'CPC', and 'lollipop-aged' are aged by default, while small is 'pristine' by definition. So we only need to decide if 'column', 'plate' and 'frozen drops' are aged or not. Usually, when an ice/component is aged, it has some tiny bumps on the edges.

[Figure]

*Figure 1 A randomly selected sample of ice crystal images from each category*

[Figure]

*Figure 2 The process of classifying ice crystals*

While we acknowledge that the 'CPC' and 'lollipop' categories may appear to be more specialized, their inclusion was intentional and aimed at understanding the microphysical processes responsible for their creation (Pasquier et al. 2023). Including these habits highlights the ability of IceDetectNet to maintain high performance on well-defined categories (i.e. column, plate), while still providing a mechanism to classify ice crystals that do not fit nicely into traditional habit classes (e.g. lollipop-aged).

We understand the critical role that a wider range of ice crystal habits, such as needles, play in mixed-phase clouds. The absence of these habits in our dataset limits the inclusion of these habits in the classification scheme and in the training and validation process of IceDetectNet. In our extended discussion (see below), we highlight the potential of fine-tuning IceDetectNet to integrate ice crystal habits that were not part of the original training set.

**L94-99:**

"For a more detailed discussion of the categorization criteria, we refer to Appendix B. However, due to data limitations, our dataset does not capture every possible basic ice habit, such as needles and rosettes. This limitation is acknowledged and further discussed in Sect.5 where we look at potential extensions to IceDetectNet. As new data containing additional ice habits become available, IceDetectNet can be updated as it is designed to incorporate these new habits through fine-tuning, ensuring the continued evolution of the model."

**L470-471:**

"As the current dataset used for the training of IceDetectNet does not include some basic habits such as needles and rosettes, we plan to adapt IceDetectNet to include these categories once additional datasets containing these habits are available. "

2. Small: why discriminate between small and irregular habits? Why not use an objective criteria for small, like measured maximum dimension? Tell the reason of having small and irregular.

Thank you for raising this point. The distinction between "small" and "irregular" ice crystal habits allows us to use shape information effectively, rather than simply classifying based on size. Small' refers to crystals that are generally too small to accurately determine their shape, but this category also includes crystals such as small 'column' ice where the shape is still recognizable. Conversely, some larger crystals may have indistinguishable shapes.

Following your suggestions, we investigated different metrics, such as equivalent size, major axis size, minor axis size, and particle area, to identify the best threshold for defining 'small' ice crystals. The major axis dimension was found to be the most reliable measure and showed the least overlap between 'small' and the rest of the categories ice in our analyses (as shown in Fig 3).

[Figure]

*Figure 3: Histogram comparing major axis sizes of 'small' ice crystals and other categories (non-small).*

We found that a threshold of 75-80 µm allows us to keep shape information for over 80% of the ice crystals (as shown in Figure 4). However, below this threshold, some small crystals can still be detected and classified, providing insight into their shapes and contributing to our understanding of ice crystal shape. If we follow your suggestion and set a rigid threshold, the information below this threshold will be completely lost. Also, those 'small' ice larger than the threshold will be fed into the IceDetectNet and then forced to be classified into a certain class, which would lead to some confusion for the IceDetectNet.

[Figure]

*Figure 4 Histogram of non-small ice crystals as a percentage of total ice crystals (small + non-small) by major axis size.*

Therefore, based on this assessment, we retain the current classification without imposing a fixed size threshold for 'small' crystals. This approach ensures that our model utilizes the full range of available data. In the future, we may consider implementing a 75-micron threshold to further refine our classification framework based on the insights gained from the current analysis.

3. Pristine: The definition of pristine, that the authors are presenting, is too broad. According to this definition, any ice crystal that is not rimed, sublimated or aggregated is classified as pristine, including polycrystalline habits or other highly complex shapes. Korolev et al. (1999) defined pristine ice as "faceted single ice crystals". The authors should adopt a similar narrower definition.

Thank you for pointing this out and we agree that our definition was not very clear before. Here we only define pristine as 'column', 'plate', 'frozen droplet' and 'small' ice crystals, as detailed in Table 1. To make this clearer we have

added a section to the appendix of our manuscript (see Question 1 **Appendix B: Detailed criteria for ice crystal classification**). Also, we added one more definition sentence in the Section 2 Data description (see below).
**L93-94:**
"Furthermore, all ice crystals in the classes 'lollipop', 'CPC' and 'irregular' are defined as aged ice instead of pristine ice (as they are not newly-produced ice) while they are still basic habit categories."

4. Aged: It is more precise to categorize ice crystals based on their physical state as "rimed" or "sublimated" rather than using the term "aged," which does not directly correlate with a specific physical transformation.

We recognize the importance of accurately categorizing ice crystals into 'rimed' or 'sublimated' states. However, the distinction between 'rimed' and 'sublimated' states is not always clear within a dataset and requires the interpretation of a researcher (see example images below). Therefore, here we use 'aged' as a merged category to avoid potential misclassifications by IceDetectNet. In doing so, it allows researchers the flexibility to perform in-depth physical analyses tailored to their specific needs. For example, a researcher can further classify the 'aged' category predicted by IceDetectNet into "riming" or "sublimated" when considering additional data such as meteorological or microphysical conditions for physical interpretation.

To help clarify these categories, we have included a sample image section in the appendix to provide a better overview of the dataset. The same as above:
**Appendix B: Detailed criteria for ice crystal classification**

5. The manuscript prompts concerns regarding the precision of manual classification processes employed. Ensuring the accuracy of training and test datasets is crucial for the success of deep learning methodologies. A case in point is the classification of a crystal in Figure 3 as CPC, despite the absence of plate-like features, and its closer resemblance to a needle structure. Given that CPC crystals suggest a transitional phase between columnar and plate growth regimes, as discussed by Pasquier et al. (2023), and needle growth typically occurs at warmer temperatures around -5°C, such misclassification could potentially lead to incorrect assumptions about the environmental conditions experienced by the crystal. To address these concerns, the authors are advised to discuss the classification criteria for each ice crystal habit more comprehensively and provide additional examples from various habit

categories. This approach would significantly enhance confidence in the accuracy of the manual labeling process.

Thanks for pointing this out and admittedly, the classification criteria for classifying ice crystals are not very clear. In response to your previous comments, we have included an additional section to the appendix (see Question 1: **Appendix B: Detailed criteria for ice crystal classification**) that outlines the criteria for classifying the ice crystals, including some randomly selected crystals as examples.

6. The manuscript introduces three distinct classification frameworks: multi-label classification, basic habit identification, and microphysical process categorization. In addition, it explores a rotated object detection algorithm for discerning aggregates and their respective components. However, the integration and mutual dependencies among these classification strategies are not clearly articulated, leaving the reader uncertain about how the results from these different schemes are interrelated. A particularly crucial element of the research is the aggregate detection capability, where the method for identifying aggregates through bounding boxes is noted to be quite effective, boasting an accuracy rate of 92% as detailed in Section 4.1.1. Conversely, Table 2 reveals a significantly lower accuracy rate of only 50.3% for the multi-label classification scheme in recognizing aggregate classes, indicating suboptimal performance in this aspect. To address these discrepancies, it is recommended that the authors consider implementing a sequential classification approach. This would entail initially employing the rotated object detection algorithm to differentiate between aggregate and non-aggregate forms, followed by the classification of the individual crystals' habits, and culminating with the identification of the microphysical processes involved.

Thank you for pointing this out and we now realize that the differences among multi-label classification, basic habit identification, and microphysical process categorization are not very clear to the readers. Thus, in the added new section in the appendix (see Question 1: **Appendix B: Detailed criteria for ice crystal classification**), we also added a diagram of how we do the hand-label in basic habit and microphysical processes and how to combine both of the information. To clarify, the 92% shown in Section 4.1.1 is classifying whether the ice is aggregated (with more than one bounding box predicted) or single (with 1 bounding box predicted only), which is for evaluating the performance of detection. However, the 50.3% in Table 2 shows how well we could classify these aggregated ice crystals in basic habits and microphysical

processes, which includes both detection (e.g. whether multiple were detected for aggregate ice) and classification (e.g. whether the basic habit is classified correctly). They are two different level tasks in difficulty and the latter one is much more difficult than the first one.

7. The authors have decided to separate the training and test data sets to different temperature regimes. The reasoning behind this is that the test and training datasets will have different characteristics. Consequently, it is challenging to assess the algorithm's generalization capability across unrepresented habits in the test dataset, such as frozen droplet, CPC, lollipop, and their aged and aggregated forms. Typically, it is advisable for both the training and test datasets to encompass all classes to ensure a comprehensive evaluation of the algorithm's performance. The authors might want to reconsider their approach regarding the selection of training and test datasets or reevaluate the defined habit classes in light of these concerns. Additionally, the representation of some classes within the training dataset, such as column aggregate, is insufficiently robust for effective training. It may be beneficial to exclude classes with inadequate sample sizes to improve the reliability and validity of the classification outcomes.

We completely agree that the description of the training, test and validation sets was confusing as well as how the training dataset was divided into the splits. With this in mind, we have renamed the test dataset to generalization dataset, explicitly described the division of the training set, and updated the main text accordingly:
**L100-106:**
"The dataset collected on November 11, 2019 (Pasquier et al., 2022b), hereafter training dataset, was used to train IceDetectNet. During the training it was divided into a training subset (comprising 80% of the data) and a validation subset (made up of the remaining 20%), using a cross-validation method (detailed introduction in Sect. 3.8.1). This validation subset serves a similar purpose as the traditional test sets used in other studies (Jaffeux et al., 2022; Xiao et al., 2019; Touloupas et al., 2020), providing an initial evaluation of the model's performance under known conditions. On the other hand, the generalization dataset was collected on a different date, April 1, 2020 (Pasquier et al., 2022b) which is not used during training but to evaluate the generalization abilities of IceDetectNet."

To clarify, here we use two different and independent datasets: a training dataset, and a generalization dataset, each of which serves different purposes and is collected during different seasons.

1. Training datasets:  We split our training data into two subsets: 80% for training and 20% for validation. The training subset helps the model learn data patterns, while the validation subset checks its effectiveness on unseen data to ensure good generalization. We also use a 5-fold cross-validation method to prevent overfitting by dividing the data set into five equal parts. Each cross-validation cycle uses four parts for training and one for validation, alternating so that each part serves as validation data once.

2. Generalization dataset: The generalization dataset is separate from the training process. It is designed to evaluate the model's ability to generalize across different environmental conditions, especially those not represented in the training dataset. This dataset was intentionally collected under significantly different conditions (from -23 to -15°C) to challenge the model with ice particle types that may not be present in the training data. We acknowledge that this dataset does not include classes such as 'lollipop' or 'CPC' due to their absence under the specific conditions and seasons of collection.

Regarding the exclusion of classes with insufficient sample sizes, the real data distribution in our current datasets reflects the actual presence of different ice shapes under specific conditions and seasons. Excluding classes due to insufficient numbers could bias the model away from these realities. Furthermore, each additional data image contributes valuable information that enhances model learning, making the inclusion of all available data critical. Recognizing the limitations of the current dataset in representing all possible ice habits, especially underrepresented or missing categories, we plan to use fine-tuning methods for upcoming new datasets, as detailed in the extended discussion section (see Question 1: **Appendix B: Detailed criteria for ice crystal classification**).

8. The performance of the training dataset is well discussed and shown in Figs. 4-7. The performance of the test dataset is only shortly discussed in Sec. 4.3. Figure 8c shows the overall performance of basic habits, microphysical processes, and all-classes but no detailed results for the different classes are shown similar the training dataset (Figs. 5 and 6).

Thanks for pointing this out. We have now included the confusion matrix plots for the generalization dataset and added the analysis on the plots as follows: **L433-456:**

"To gain further insights into IceDetectNet's performance in each ice category, here we analyzed the confusion matrices (mean of 5 models) for basic habit classification (Fig.9) and microphysical processes classification (Fig.10) for the generalization dataset as well.

IceDetectNet achieved an overall accuracy of 81% for the basic habit categories (Fig.9) The confusion matrix shows that IceDetectNet still performed better for the ice categories that comprise a large fraction of the dataset, like 'small' (precision of 90%). However, among these, 'column' classification performance had a large performance drop (18% decrease in precision compared to the training dataset), and its two main misprediction sources were 'irregular' and 'plate', which almost represent all the mispredictions (29%). This could be due to the data distribution shift from 'column' to 'plate'. Under the general decrease trend among all categories, 'irregular' surprisingly has a 10% increase in precision. The main misprediction of 'irregular' comes from 'column' in both the training dataset and generalization data, which could be the reason that IceDetectNet learned many column features in the training dataset and thus distributed higher weights on these column features. While the number of 'column' is much less in the generalization dataset and thus leads to better performance in classifying 'irregular'. For the missing categories like 'lollipop' and 'CPC' that had zero actual occurrences, IceDetectNet still predicted 113 ice as 'lollipop' and 50 as 'CPC', with most misclassifications as 'irregular'. This problem is likely due to the model's handling of sparse data and its tendency to fit 'irregular' into these less common categories, since 'irregular' learned the most complex features since any unrecognizable shape is 'irregular'.

For the microphysical processes category (Fig.10), IceDetectNet achieved an overall accuracy of 73% (with a 9% drop compared to the training dataset). The model still performed well in identifying 'pristine' ice crystals (82%). In contrast, it shows a better performance in predicting 'aggregate' (17% higher than in the training dataset) and 'aged-aggregate' (7% higher than in the training dataset) ice crystals. This could be due to the changes in the data distribution, especially the changes in the aggregate fraction from 12% in the training dataset to 37.7% in the generalization dataset, which further emphasizes the importance of the balance of the dataset. After checking the main source of misprediction, we can see that underdetection still plays an important role, for example, the main source of misprediction of 'aggregate' is 'pristine', which is a typical misprediction problem."

**Confusion matrix**

|  | Column | CPC | Droplet | Irregular | Lollipop | Plate | Small | Sum_actual |
|---|---|---|---|---|---|---|---|---|
| **Column** | 1751.0 71.64% |  | 0.2 0.87% | 787.4 11.50% |  | 242.4 13.77% | 55.8 1.65% | 2836 61.72% |
| **CPC** | 17.0 0.70% |  |  | 31.6 0.46% |  | 2.2 0.12% | 1.8 0.05% | 52 0.00% |
| **Droplet** | 7.0 0.29% |  | 13.4 58.26% | 29.6 0.43% |  | 19.0 1.08% | 1.8 0.05% | 70 18.93% |
| **Irregular** | 418.6 17.13% |  | 4.8 20.87% | 5627.2 82.16% |  | 242.6 13.78% | 227.2 6.71% | 6520 86.30% |
| **Lollipop** | 7.4 0.30% |  |  | 84.2 1.23% |  | 2.8 0.16% | 18.6 0.55% | 113 0.00% |
| **Plate** | 229.6 9.39% |  | 4.6 20.00% | 147.4 2.15% |  | 1237.4 70.31% | 27.6 0.82% | 1646 75.15% |
| **Small** | 13.4 0.55% |  |  | 142.0 2.07% |  | 13.6 0.77% | 3051.6 90.17% | 3220 94.75% |
| **Sum_predicted** | 2444 71.64% | 0 0.00% | 23 58.26% | 6849 82.16% | 0 0.00% | 1760 70.31% | 3384 90.17% | 14460 80.77% |

Predicted (y-axis) / Actual (x-axis)

*Figure 5 Similar confusion matrix as Fig. 5 in the manuscript, but for generalization dataset. Class 'CPC' and 'Lollipop' are missing in the dataset and thus are 0 in its corresponding column.*

*Figure 6 Similar confusion matrix as Fig. 6 in manuscript, but for generalization dataset*

**9.** Table 2: since the classes are highly imbalanced, overall accuracy is not necessarily the best performance metric. Other performance metrics, such as balanced accuracy or F1-score could work better.

This is a good point and we have now included the F1 score for each class, the mean F1 score and its standard deviation in Table 2 and Table 3 for a more comprehensive overview of the model performance.

**Table 2.** Overall accuracy of the multi-label, basic habit and microphysical processes ice classification. The table displays the overall accuracy values for each of the five models, along with the mean and standard deviation (std) values (all values are reported in percentages). The validation set is broken down into 'aggregate' (agg) and 'non-aggregate' (non-agg) subsets.

1 2 3 4 5 mean std All data (19-class) Non-agg (10-class) 82.5 81.3 81.6 81.2 83.5 82.0 1.0 Agg (9-class) 46.1 53.7 54.4 47.0 50.5 50.3 3.8 Non-agg (7-class) 89.4 90.0 89.4 88.8 90.7 89.7 0.7 Agg (6-class) 71.7 76.1 72.6 70.7 71.3 72.5 2.1 Non-agg (2-class) 85.6 84.0 84.6 84.7 86.2 85.0 0.9 Agg (2-class) 48.9 55.0 56.1 48.3 52.4 52.1 3.5

| | | 1 | 2 | 3 | 4 | 5 | mean OA | mean F1 score | std-OA | std-F1 |
|---|---|---|---|---|---|---|---|---|---|---|
| **Multi-label** | **All data (19-class)** | 78.1 | 78.0 | 78.3 | 77.0 | 79.4 | **78.2** | **54.9** | 0.9 | 1.9 |
| | Non-agg (10-class) | 82.5 | 81.3 | 81.3 | 83.5 | 82.0 | 82.1 | 71.7 | 0.9 | 1.6 |
| | Agg (9-class) | 46.1 | 53.7 | 54.4 | 47.0 | 50.5 | 50.3 | 41.4 | 3.8 | 5.9 |
| **Basic Habit** | **All data (7-class)** | 86.5 | 86.5 | 86.3 | 85.6 | 87.2 | **86.4** | **78.8** | 0.6 | 1.3 |
| | Non-agg (7-class) | 89.4 | 90.0 | 89.4 | 88.8 | 90.7 | 89.7 | 81.8 | 0.7 | 1.3 |
| | Agg (6-class) | 71.7 | 76.1 | 72.6 | 70.7 | 71.3 | 72.5 | 58.5 | 2.1 | 3.4 |
| **Microphysical Processes** | **All data (4-class)** | 81.4 | 82.0 | 81.3 | 80.8 | 82.6 | **81.6** | **66.9** | 0.7 | 1.1 |
| | Non-agg (2-class) | 85.6 | 84.0 | 84.6 | 84.7 | 86.2 | 85.0 | 84.3 | 0.9 | 1.0 |
| | Agg (2-class) | 48.9 | 55.0 | 56.1 | 48.3 | 52.4 | 52.1 | 62.3 | 3.5 | 2.8 |

**Table 3.** Overall accuracy of the multi-label, basic habit and microphysical processes ice classification. The table displays the overall accuracy values for each of the five models, along with the mean and standard deviation (std) values (all values are reported in percentages). The generalization dataset is broken down into 'aggregate' and 'non-aggregate' subsets.

| | | 1 | 2 | 3 | 4 | 5 | mean OA | mean F1 score | std-OA | std-F1 |
|---|---|---|---|---|---|---|---|---|---|---|
| **Multi-label** | **All data (14-class)** | 67.5 | 66.7 | 67.2 | 67.5 | 68.3 | **67.5** | **48.5** | **0.6** | **1.6** |
| | Non-agg (8-class) | 72.8 | 73.3 | 71.2 | 74.5 | 73.2 | 73.0 | 58.3 | 1.2 | 2.0 |
| | Agg (6-class) | 46.3 | 50.0 | 53.1 | 47.7 | 51.2 | 49.7 | 45.4 | 2.7 | 1.4 |
| **Basic habit** | **All data (7-class)** | 81.6 | 79.9 | 79.6 | 80.7 | 81.3 | **80.6** | **68.7** | **0.8** | **0.6** |
| | Non-agg (5-class) | 86.4 | 85.9 | 85.4 | 84.8 | 86.7 | 85.9 | 70.3 | 0.7 | 0.8 |
| | Agg (5-class) | 64.7 | 69.1 | 65.3 | 69.6 | 71.7 | 68.1 | 61.0 | 2.9 | 1.7 |
| **Microphysical processes** | **All data (4-class)** | 72.8 | 72.3 | 72.5 | 72.9 | 73.4 | **72.77** | **64.8** | **0.4** | **0.3** |
| | Non-agg (2-class) | 78.5 | 78.0 | 77.4 | 78.1 | 77.3 | 77.8 | 74.6 | 0.5 | 0.8 |
| | Agg (2-class) | 45.2 | 51.1 | 50.9 | 46.6 | 48.4 | 48.4 | 67.9 | 2.6 | 0.6 |

10. Line 4: can you change "density" to "effective density"

Changed, thanks.

11. Lines 30-32: it is also possible that ice crystals change environment in convective systems or by precipitation, which can lead to formation of mixed habits.

Rephrased:
**L29-32:**
"meteorological …... The change in the ambient environment, such as in a convective system, leads to a complex
basic habit such as columns on capped columns (CPCs) (observed by (Pasquier et al., 2023))."

12. Lines 35-36: Schmitt & Heymsfield (2014) defined a complexity parameter to discriminate between single crystals and aggregates. This work could be also cited.

Cited, thanks

13. Lines 49-52: There are multiple studies investigating the habits (pristine or single habits vs irregular habits) of mixed-phase or cirrus clouds besides that of Korolev et al. (1999). It is advised to give a broader overview of these observations.

We added

**L56-60:**

"Moreover, in stratiform clouds, Korolev et al. (2000) found that 84% of the ice crystals are irregular, which includes everything except needles and dendrites. These irregular ice crystals would either be aged or aggregated by our definition (refer Sect. 2 and Sect. B)."

However, it is important to clarify that the primary focus here is to highlight the potential loss of information when basic habits and microphysical processes are not considered simultaneously.

14. Line 273: How did the different folds perform?

We have reported the standard deviations for the model's performance across the 5 folds in Table 2 and Table 3. The relatively small OA standard deviations (below 1%) indicate that the performance of the model is consistent across all folds. Therefore, for conciseness and clarity, we have not included the performance of each fold everywhere, as the variations are minimal.

**Reference:**

Cunningham R M. Analysis of particle spectral data from optical array (PMS) 1D and 2D sensors[C]//AMS 4th Symposium on Meteorol. Obs. Instruments. 1978: 10-14.

Duroure C, Larsen H R, Isaka H, et al. 2D image population analysis[J]. Atmospheric research, 1994, 34(1-4): 195-205.

Korolev A, Isaac G A, Hallett J. Ice particle habits in stratiform clouds[J]. Quarterly Journal of the Royal Meteorological Society, 2000, 126(569): 2873-2902.

Pasquier, J. T., Henneberger, J., Ramelli, F., Lauber, A., David, R. O., Wieder, J., Carlsen, T., Gierens, R., Maturilli, M., and Lohmann, U.:
Conditions favorable for secondary ice production in Arctic mixed-phase clouds, Atmospheric Chemistry and Physics, 22, 15 579–15 601,2022b.

Pasquier, J. T., Henneberger, J., Korolev, A., Ramelli, F., Wieder, J., Lauber, A., Li, G., David, R. O., Carlsen, T., Gierens, R., et al.:
Understanding the history of two complex ice crystal habits deduced from a holographic imager, Geophysical Research Letters, 50, e2022GL100 247, 2023

Schmitt C G, Heymsfield A J. Observational quantification of the separation of simple and complex atmospheric ice particles[J]. Geophysical Research Letters, 2014, 41(4): 1301-1307.

---

## Referee Report (RR1)

**Comments on "IceDetectNet: A rotated object detection algorithm for classifying components of aggregated ice crystals with a multi-label classification scheme" by Huiying Zhang et al. (https://doi.org/10.5194/egusphere-2023-2770)**

The manuscript proposes a novel ice crystal classification method that employs a two-step process. Initially, a rotated object detection algorithm (IceDetectNet) is applied to categorize the individual component of aggregated ice crystals. Secondly, a multi-label classification scheme is applied, whereby the ice crystals are categorized according to their basic habits, as well as the physical processes that modified them. The IceDetectNet algorithm was trained and tested on data sets obtained from a holographic imager.

The proposed classification method represents a step forward and an improvement compared to traditional machine learning approaches. However, in its current form it is subject to a number of limitations.

It is acknowledged that the authors have addressed the majority of the comments that were raised during the previous reviews. Particular attention is paid to the first and second reviews from Reviewer 1 in which concerns about the accurate labeling of the ice crystal shapes were raised. Within the first and second revision the authors provided a slightly improved version of the main text and added Appendix B. However, the figures given in Fig. 2B still raise questions. In case that the individual crystal images are truly randomly selected, the figure provides evidence of potential false classification: Needles might be falsely classified as columns. In this regard I do agree with Reviewer 1 (1st and 2nd round) that several crystals in the "column-aged", "CPC", and the "column-aged-aggregate" category appear to be needles. This raises the question of weather the appearance of the needles is deliberate (e.g., selected as columns) or a miss-classification during the training or the application.

A further critical aspect are the chosen shape categories. Although the selection of ice crystal shapes and physical processes was justified based on the training data set, the selection is however unconventional. For example the distinction between "small" vs "irregular", as small refers to size and irregular refers to shape. Strictly speaking, it is a comparison of size and shape. Also the choice to differentiate between "pristine" and "aged" is an unlucky selection, in my opinion. Here, pristine refers to shape, while aged refers to a temporal development. While this distinction and the categorization of ice crystals in the manuscript may be applicable to this specific data-set from Ny-Ålesund, it limits the applicability to other data sets and makes it challenging to compare results obtained from IceDetectNet with those from the existing literature.

In this regard, I acknowledge the discussion of the limitations of the current version of IceDetectNet in Section 5. However, it would be fair to the reader of the manuscript / paper to explicitly mention these limitations in the Introduction and the Conclusion. By "limitations" I refer to the restricted shapes in the training data set as well as the unconventional selection of ice crystal shapes and microphysical classes.

The manuscript can be considered for publication, when the limitations from above are explicitly mentioned in the introduction and the conclusion part.

Minor comments:

Line 505 / 507: FigB2 is called before FigB1, so I suggested to change the labeling to keep the order.

---

## Author Response (AR2)

**Authors' comments to Anonymous Referee #1**

We would like to thank the reviewer for the thorough review of our manuscript and insightful feedback. These comments have significantly improved the quality of our work. In the following sections, we present the reviewer's comments (in black), our responses (in red), and the changes made in the revised manuscript (in blue). Please note that all line numbers in our responses correspond to those in the revised manuscript.
* * *
**Comments:**

The authors have adequately addressed the majority of the points raised by the reviewers and improved the description of the training and generalization datasets. Additionally, the discussion on the performance of the generalization dataset is now more comprehensive, and the rationale behind tailoring the ice habit categories for the available dataset is acceptable.

However, the addition of the Appendix B (Detailed criteria for ice crystal classification) has not sufficiently resolved the concerns related to the manual labeling of ice crystal habits. In fact, the sample ice crystal images presented in the appendix have raised further issues. The problematic habit categories include Column-aggregate, Column-aged, CPC-aggregate, and CPC. Many of the "columns" in these categories appear to be needles. According to Kikuchi et al. (2013), needles are defined as "crystals shaped like needles, with tops shaped like knife-edges."

In the CPC and CPC-aggregate category many of the example crystals are missing the plate "P" section of the crystal. Following the Kikuchi diagram the crystals classified as CPC by the authors fall in the categories C1a (needle) or C1b (needle bundle). Only in the CPC-aggregate category plate growth is evident for some crystals. Following Fig. 2 of Appendix B, the classification criteria for columns and CPCs become unclear, particularly the distinction between "rectangle-based" and "hexagonal-based." It is advised that these steps are to be revised.

Thank you for your comments. We understand your concerns regarding the manual labeling of ice crystal habits. As mentioned in Section 2 of the manuscript, here we use the habit classes and datasets from Pasquier et al. (2022a, 2022b, 2023), which were published on Zenodo (https://zenodo.org/records/7402285). These published datasets include ice crystals that were manually classified into different ice categories. Although we agree with the reviewer that in some instances there may have been some misclassification of the ice crystals, an improvement to the

classified data set is not the aim of this study. The aim of this study is to introduce a novel algorithm (IceDetectNet) for classifying ice crystals down to the aggregated component level. With this in mind, we are confident that these misclassifications do not influence the ability of IceDetectNet's implementation to identify and classify ice crystals at the component level.

Following your comments, we made the following changes to the main text:

1. Column-aggregate and CPC-aggregate:
To clarify the classification of aggregated ice, we have revised the text and the caption of the Fig. B2 as follows:

L511-512
"In multi-label classification, only the largest visually identified component of the aggregate is classified, without drawing a bounding box. The classified basic habit of this component will represent the basic habit of the whole aggregate. "

Fig. B2 Caption:
A randomly selected sample of ice crystal images from each category based on the multi-label classification scheme

Thus, if some components within the aggregate do not fit into the basic habit category of aggregated ice, it is still correct based on our definition. For example, if a 'CPC' ice is aggregated with a 'Column', as long as the largest component of the aggregate is the CPC, then the label of the aggregate should still be 'CPC-aggregate'.

2. CPC and Column:
We agree that some of the CPCs are missing the "P" portion of the crystal and fall into the C1a and C1b categories in the Kikuchi diagram. Nevertheless, in our dataset, these ice crystals are classified as CPC following the argumentation of Pasquier et al. 2023, which considers the growth history of these ice particles, covering both the column and plate temperature regime (See Section 3 Results, especially Section 3.1. Observations of Columns on Capped-Column). Therefore, the classification is working correctly, even if the crystals could have been originally classified as C1a and C1b.

To address the confusion between 'CPC' and 'Column' caused by rectangular and hexagonal-based features, we have added further explanations to differentiate the ice habits more clearly, as follows:

L516-518
"Rectangular-shaped ice crystals (with 4 distinct edges) are classified as 'columns', whereas rectangular-shaped ice crystals with multiple branches at the end of the maximum dimension are labeled 'CPC-aged. Note, that the CPC-aged categories also include needle bundles with missing plate sections. Hexagonal crystals (with 6 distinct edges) are classified based on their aspect ratio, where a high aspect ratio indicates a 'column,' and a low aspect ratio a 'plate'."

L96-97
"However, due to data limitations, our dataset does not capture every possible basic ice habit, such as needles and rosettes, but the existence of these ice habits is well acknowledged (Kikuchi et al., 2013)."

The reason why both 'rectangular-based' and 'hexagonal-based' could fall into the 'column' category is due to the different viewing angles of the ice particles (as all images are 2-D). Here you can see an example of each:

[Figure]

*Figure 1 'Column' ice with hexagonal-based structure (6 distinct edges)*

[Figure]

*Figure 2 'Column' ice with rectangle-based structure (4 distinct edges)*

We hope that these clarifications and revisions address your concerns. Thank you again for your valuable feedback, which has helped to improve the quality of our manuscript.

**References**

Kikuchi K, Kameda T, Higuchi K, et al. A global classification of snow crystals, ice crystals, and solid precipitation based on observations from middle latitudes to polar regions[J]. Atmospheric research, 2013, 132: 460-472.

Pasquier J T, David R O, Freitas G, et al. The Ny-Ålesund aerosol cloud experiment (nascent): Overview and first results[J]. Bulletin of the American Meteorological Society, 2022, 103(11): E2533-E2558.

Pasquier, J. T., Henneberger, J., Ramelli, F., Lauber, A., David, R. O., Wieder, J., Carlsen, T., Gierens, R., Maturilli, M., and Lohmann, U.:
Conditions favorable for secondary ice production in Arctic mixed-phase clouds, Atmospheric Chemistry and Physics, 22, 15 579–15 601,2022b.

Pasquier, J. T., Henneberger, J., Korolev, A., Ramelli, F., Wieder, J., Lauber, A., Li, G., David, R. O., Carlsen, T., Gierens, R., et al.:
Understanding the history of two complex ice crystal habits deduced from a holographic imager, Geophysical Research Letters, 50, e2022GL100 247, 2023a

---

## Author Response (AR3)

**Authors' comments to Anonymous Referee #1**

We would like to thank the reviewer for the thorough review of our manuscript and insightful feedback. These comments have significantly improved the quality of our work. In the following sections, we present the reviewer's comments (in black), our responses (in red), and the changes made in the revised manuscript (in blue). Please note that all line numbers in our responses correspond to those in the revised manuscript.
* * *
**Overall comments:**

The manuscript proposes a novel ice crystal classification method that employs a two-step process. Initially, a rotated object detection algorithm (IceDetectNet) is applied to categorize the individual component of aggregated ice crystals. Secondly, a multi-label classification scheme is applied, whereby the ice crystals are categorized according to their basic habits, as well as the physical processes that modified them. The IceDetectNet algorithm was trained and tested on data sets obtained from a holographic imager. The proposed classification method represents a step forward and an improvement compared to traditional machine learning approaches. However, in its current form it is subject to a number of limitations.

1. A further critical aspect are the chosen shape categories. Although the selection of ice crystal shapes and physical processes was justified based on the training data set, the selection is however unconventional. For example the distinction between "small" vs "irregular", as small refers to size and irregular refers to shape. Strictly speaking, it is a comparison of size and shape.

   We agree that the distinction between "small" and "irregular" ice crystals is based on a combination of size and shape information rather than a strict size versus shape comparison. "Small" refers to crystals that are often too limited in pixel number to accurately determine their shape. In contrast, "irregular" refers to larger crystals with indistinguishable shapes due to more complex structural features. As stated in the first round of responses (see the response to comment #2 Reviewer #1"), setting a rigid threshold for the 'small' category would result in a loss of valuable shape information for crystals below this size (Fig. 1). Certain shapes, such as 'column', require fewer pixels for accurate recognition, while more complex structures may require additional pixel detail. Thus, keeping these two categories distinct allows greater flexibility in recognizing and classifying ice crystals of varying structural complexity, ensuring that the model captures both size and shape variation without being

forced into rigid thresholds. Also, these categories are based on the ice classification in Pasquier et al. (2022a, 2022b, 2023).

[Figure]

*Figure 1: Histogram comparing major axis sizes of 'small' ice crystals and other categories (non-small).*

2. Also the choice to differentiate between "pristine" and "aged" is an unlucky selection, in my opinion. Here, pristine refers to shape, while aged refers to a temporal development. While this distinction and the categorization of ice crystals in the manuscript may be applicable to this specific data-set from Ny-Ålesund, it limits the applicability to other data sets and makes it challenging to compare results obtained from IceDetectNet with those from the existing literature.

Thanks for your insightful comment. Both the terms "pristine" and "aged" as we use them refer to differences in shape, with "pristine" indicating that the ice crystal has just formed and maintained a regular and easily identifiable shape. For example, a pristine column might appear as a well-defined rectangular shape. On the other hand, 'aged' includes crystals that have undergone various microphysical processes, such as riming or sublimation, resulting in more irregular or complex shape features.

In response to your comment, we have revised the definition of 'Pristine' in Table 1 in the manuscript:

Pristine: Ice crystals with an easily identifiable shape that have not undergone any microphysical processes.

3. In this regard, I acknowledge the discussion of the limitations of the current version of IceDetectNet in Section 5. However, it would be fair to the reader of the manuscript / paper to explicitly mention these limitations in the Introduction and the Conclusion. By "limitations" I refer to the restricted shapes in the training data set as well as the unconventional selection of ice crystal shapes and microphysical classes. The manuscript can be considered for publication, when the limitations from above are explicitly mentioned in the introduction and the conclusion part.

Thank you for pointing out that the limitations of the IceDetectNet should also be included in the introduction and conclusion. We have added them as follows:

**L73-75:**
However, like all supervised learning methods, our approach is limited to the ice categories present in the training dataset, limiting its applicability until the model is fine-tuned on a new dataset.
**L494-498:**
However, the ice categories used in this study are specific to environmental and microphysical conditions present during the collection of the training data. In addition, the distinction between 'small' and 'irregular' ice categories combines both size and shape information, making it difficult to be classified. While these categories are appropriate for the current dataset, they may pose challenges when applying IceDetectNet to other datasets or comparing results with existing studies. However, adding or refining categories can be easily achieved through model fine-tuning.

**Minor comments:** Line 505 / 507: FigB2 is called before FigB1, so I suggested to change the labeling to keep the order.

Fixed, thanks!

**References**
Pasquier J T, David R O, Freitas G, et al. The Ny-Ålesund aerosol cloud experiment (nascent): Overview and first results[J]. Bulletin of the American Meteorological Society, 2022, 103(11): E2533-E2558.

Pasquier, J. T., Henneberger, J., Ramelli, F., Lauber, A., David, R. O., Wieder, J., Carlsen, T., Gierens, R., Maturilli, M., and Lohmann, U.:
Conditions favorable for secondary ice production in Arctic mixed-phase clouds, Atmospheric Chemistry and Physics, 22, 15 579–15 601,2022b.

Pasquier, J. T., Henneberger, J., Korolev, A., Ramelli, F., Wieder, J., Lauber, A., Li, G., David, R. O., Carlsen, T., Gierens, R., et al.:
Understanding the history of two complex ice crystal habits deduced from a holographic imager, Geophysical Research Letters, 50, e2022GL100 247, 2023a